EMBO
Molecular Medicine

# Cooperation of LIM domain-binding 2 (LDB2) with EGR in the pathogenesis of schizophrenia

Tetsuo Ohnishi[1,*,†] , Yuji Kiyama[2,3,†] , Fumiko Arima-Yoshida[2,4,†], Mitsutaka Kadota[5,†] ,
Tomoe Ichikawa[6,7], Kazuyuki Yamada[8,9], Akiko Watanabe[1], Hisako Ohba[1], Kaori Tanaka[5],
Akihiro Nakaya[1,10], Yasue Horiuchi[6] , Yoshimi Iwayama[1,9], Manabu Toyoshima[1] , Itone Ogawa[2],
Chie Shimamoto-Mitsuyama[1] , Motoko Maekawa[1], Shabeesh Balan[1] , Makoto Arai[6],
Mitsuhiro Miyashita[6], Kazuya Toriumi[6] , Yayoi Nozaki[1], Rumi Kurokawa[9], Kazuhiro Suzuki[6],
Akane Yoshikawa[6], Tomoko Toyota[1] , Toshihiko Hosoya[9,11] , Hiroyuki Okuno[3] , Haruhiko Bito[12] ,
Masanari Itokawa[6] , Shigehiro Kuraku[5], Toshiya Manabe[2,**] & Takeo Yoshikawa[1,***]

## Abstract

Genomic defects with large effect size can help elucidate unknown pathologic architecture of mental disorders. We previously reported on a patient with schizophrenia and a balanced translocation between chromosomes 4 and 13 and found that the breakpoint within chromosome 4 is located near the *LDB2* gene. We show here that *Ldb2* knockout (KO) mice displayed multiple deficits relevant to mental disorders. In particular, *Ldb2* KO mice exhibited deficits in the fear-conditioning paradigm. Analysis of the amygdala suggested that dysregulation of synaptic activities controlled by the immediate early gene *Arc* is involved in the phenotypes. We show that LDB2 forms protein complexes with known transcription factors. Consistently, ChIP-seq analyses indicated that LDB2 binds to > 10,000 genomic sites in human neurospheres. We found that many of those sites, including the promoter region of *ARC*, are occupied by EGR transcription factors. Our previous study showed an association of the *EGR* family genes with schizophrenia. Collectively, the findings suggest that dysregulation in the gene expression controlled by the LDB2-EGR axis underlies a pathogenesis of subset of mental disorders.

Keywords  amygdala; balanced chromosomal translocation; behavior; ChIP-seq; knockout mouse
Subject Categories  Molecular Biology of Disease; Neuroscience

## Introduction

Schizophrenia, a chronic and debilitating mental disorder, is characterized by a variety of symptoms, including delusions, hallucinations, affective flattening, and cognitive deficits. To date, numerous "risk" genes have been reported by genome-wide association studies (GWAS; Bray & O'Donovan, 2019; Dennison *et al*, 2019) and exome sequencing (Bray & O'Donovan, 2019). However, the effect size of most of these genes is small, and it remains unclear how each genetic variation constitutes an integrative pathogenic architecture. On the other hand, a causal relationship is perspicuous in exceptional cases with gross chromosomal abnormalities. For instance, a large Scottish pedigree with multiple family members diagnosed with psychiatric disorders, including schizophrenia, bipolar disorder, and recurrent depression, has been described in the literature

1  Laboratory for Molecular Psychiatry, RIKEN Center for Brain Science, Wako, Japan
2  Division of Neuronal Network, Institute of Medical Science, the University of Tokyo, Tokyo, Japan
3  Laboratory of Biochemistry and Molecular Biology, Graduate School of Medical and Dental Sciences, Kagoshima University, Kagoshima, Japan
4  Institute of Clinical Medicine and Research, The Jikei University School of Medicine, Tokyo, Japan
5  Laboratory for Phyloinformatics, RIKEN Center for Biosystems Dynamics Research, Kobe, Japan
6  Schizophrenia Research Project, Tokyo Metropolitan Institute of Medical Sciences, Shizuoka, Japan
7  Department of Infection Control Science, Meiji Pharmaceutical University, Kiyose, Japan
8  School of Management, Shizuoka Sangyo University, Iwata, Japan
9  RIKEN, Wako, Japan
10  Department of Computational Biology and Medical Sciences, Graduate School of Frontier Sciences, The University of Tokyo, Tokyo, Japan
11  Biomedical Business Center, RICOH Company, LTD, Kawasaki, Japan
12  Department of Neurochemistry, the University of Tokyo, Graduate School of Medicine, Tokyo, Japan
   *Corresponding author Lead contact . Tel: +81 48 467 5946; Fax: +81 48 467 5946; E-mail: tetsuo.ohnishi@riken.jp
   **Corresponding author. Tel: +81 3 5449 5517; Fax: +81 3 5449 5794; E-mail: tmanabe-tky@umin.ac.jp
   ***Corresponding author. Tel: +81 48 467 5968; Fax: +81 48 467 7462; E-mail: takeo.yoshikawa@riken.jp
   †These authors contributed equally to this work

(Blackwood *et al*, 2001). Importantly, most of the subjects who developed psychiatric symptoms are shown to harbor the balanced translocation t(1;11)(q42.1;q14.3) (Blackwood *et al*, 2001). Since this translocation directly disrupts the protein coding gene termed disrupted in schizophrenia 1 (*DISC1*) in chromosome 1, researchers have pursued the pathological mechanism caused by the disruption of *DISC1* (Porteous *et al*, 2006; Mackie *et al*, 2007; Korth, 2009; Tomoda *et al*, 2017).

In 2004, our group reported a patient with schizophrenia who harbored a balanced chromosomal translocation t(4;13)(p16.1; q21.31) (Itokawa *et al*, 2004). No individuals with psychiatric symptoms or with the same chromosomal translocation were found within the second-degree relatives of the proband, thereby supporting that the translocation was causal for the proband (Itokawa *et al*, 2004). Subsequently, we set out to determine the exact breakpoints on chromosomes 4 and 13 by using next-generation DNA-sequencing analysis in combination with fluorescent *in situ* hybridization (FISH) experiments (Horiuchi *et al*, 2020). The breakpoint on chromosome 13 is within the so-called "gene desert" interval where no known genes have been mapped. The breakpoint on chromosome 4 is mapped to the 32.6-kbp upstream region of a gene encoding a putative transcription regulator lacking a DNA-binding domain, LIM domain-binding 2 (*LDB2*), also called *CLIM1* (Appendix Fig S1). By reanalyzing the NGS data (Horiuchi *et al*, 2020) carefully, we found that the proband's genome harbors large deletions in two chromosomes: chromosome 6 (10,657,982–10,660,876; 2,885 bp) and chromosome X (55,702,373–55,709,904; 7,447 bp). Due to unavailability of genomic DNAs from the proband's patients, we were not able to determine whether these deletions were *de novo* or inherited from one of the proband's parents, who manifested no psychiatric symptoms. However, no known genes are mapped to these two deleted regions. Moreover, to our best knowledge, no reports have identified these regions as susceptibility loci for schizophrenia or bipolar disorder. Although the breakpoint does not disrupt *LDB2*, *LDB2* and the breakpoint are located on the same topologically associating domain (TAD) region (Szalaj & Plewczynski, 2018; Appendix Fig S1) in the human brain (adult dorsolateral prefrontal cortex; PsychEncode; Wang *et al*, 2018). Hence, the chromosomal break should disrupt the TAD organization, in turn, affecting the interaction between the regulatory elements of *LDB2* and the gene expression. Schizophrenia and bipolar disorder share genetic risk elements. Intriguingly Horiuchi *et al* (2020) detected rare missense variants (T83N and P170L) in bipolar patients, supporting a common role of *LDB2* across mental disorders.

Mammals have a close homolog to LDB2, namely LDB1 (Matthews & Visvader, 2003; Love *et al*, 2014), and many studies have reported the role of LDB1 in multiple biological processes (Love *et al*, 2014; Costello *et al*, 2015; Ediger *et al*, 2017; de Melo *et al*, 2018; Kinare *et al*, 2019). In contrast, little is known about the biological functions of LDB2, especially in the brain, while the protein has been regarded as a layer V-specific expression marker in the cerebral cortex (Bulchand *et al*, 2003; Molyneaux *et al*, 2007). In the current study, we aimed to clarify the causal relationship between the *LDB2* deficiency and the pathogenesis of mental disorders by phenotyping *Ldb2* knockout mice and harnessing manifold approaches including behavioral, electrophysiological, biochemical, and ChIP (chromatin immunoprecipitation)-seq analyses. The present study reveals a major molecular pathway that leverages *LDB2* and immediate early genes, in the pathogenesis of mental disorders including schizophrenia and bipolar disorder.

## Results

### LDB2 is expressed in neurons of restricted regions in the brain

We first attempted to examine expression of the *LDB2* gene in the proband's patient. As a biological sample from the patient, only Epstein–Barr virus-transformed lymphoblastoid cells were available. Since the *LDB2* transcripts were not detected in those cells, we established iPS cells from those cells (LiPS; Toyoshima *et al*, 2016) from the patient and a healthy control (39 y.o, male). We observed reduced *LDB2* expression in the patient compared to the control subject in iPS cells and iPS cell-derived neurospheres (Appendix Fig S2), suggesting the chromosomal break in the patient negatively affected expression of *LDB2*.

LDB2 is a protein that lacks known DNA-binding domains but has a putative self-dimerization domain, a nuclear-localization signal, and a Lim (L̲in11, I̲sl1, and M̲ec3)-binding domain that potentially mediates binding to the self and other proteins through the Lim domain (Fig 1A). Since basic information on the LDB2 protein was limited, we first examined the tissue distribution of LDB2 in a mouse with a self-made antibody raised against the N-terminal portion of LDB2 (Fig 1 A and B) using rabbits. This region is divergent between LDB1 and LDB2 (Fig 1B), minimizing the possibility of cross-reaction of the antibody to LDB1. A Western blot analysis showed a tissue-specific and brain region-specific expression of LDB2 with apparent molecular masses of 48 and 35 kDa (Fig 1C). The lower band likely corresponds to a splice variant that lacks the C-terminal Lim-interaction domain (Tran *et al*, 2006; Fig 1A, bottom). These two bands disappeared in the tissue from

---

**Figure 1. Expression of Ldb2 in the specific regions in the mouse brains.**

A   Schematic representation of the human LDB2 protein. Positions for the variants found in bipolar patients are also shown. The shorter isoform predicted from the splice variant (Tran *et al*, 2006) is shown below.

B   Multiple alignments of the N-terminal regions of the human and mouse LDB family. Please note that the N-terminal regions are highly diverged between LDB1 and LDB2. The bar indicated the position for the peptide for antibody production.

C   Tissue distribution of the LDB2 protein. Representative Western blot image is shown. Yellow asterisks indicate uncharacterized bands.

D   Expression of Ldb2 in the cerebral cortex. DAPI-stained nuclei visualized (right) in the cerebral cortex.

E   Expression of LDB2 in the amygdala. LA; lateral amygdala, BLA; basolateral amygdala, Ce; central nucleus of amygdala.

F, G   Expression of LDB2 in neurons. The merged images obtained with LDB2 (stained in green) and NeuN (stained in red) antibodies are shown for cerebral cortex layers I–VI and the amygdala (F). Please note that most of LDB2-positive cells are NeuN-positive. (G) NeuN-positive cells/LDB2-positive cells (left) and LDB2-positive cells/NeuN-positive cells were counted in cerebral cortex layers I–VI. Data are shown as means ± SD (10 independent view fields/group).

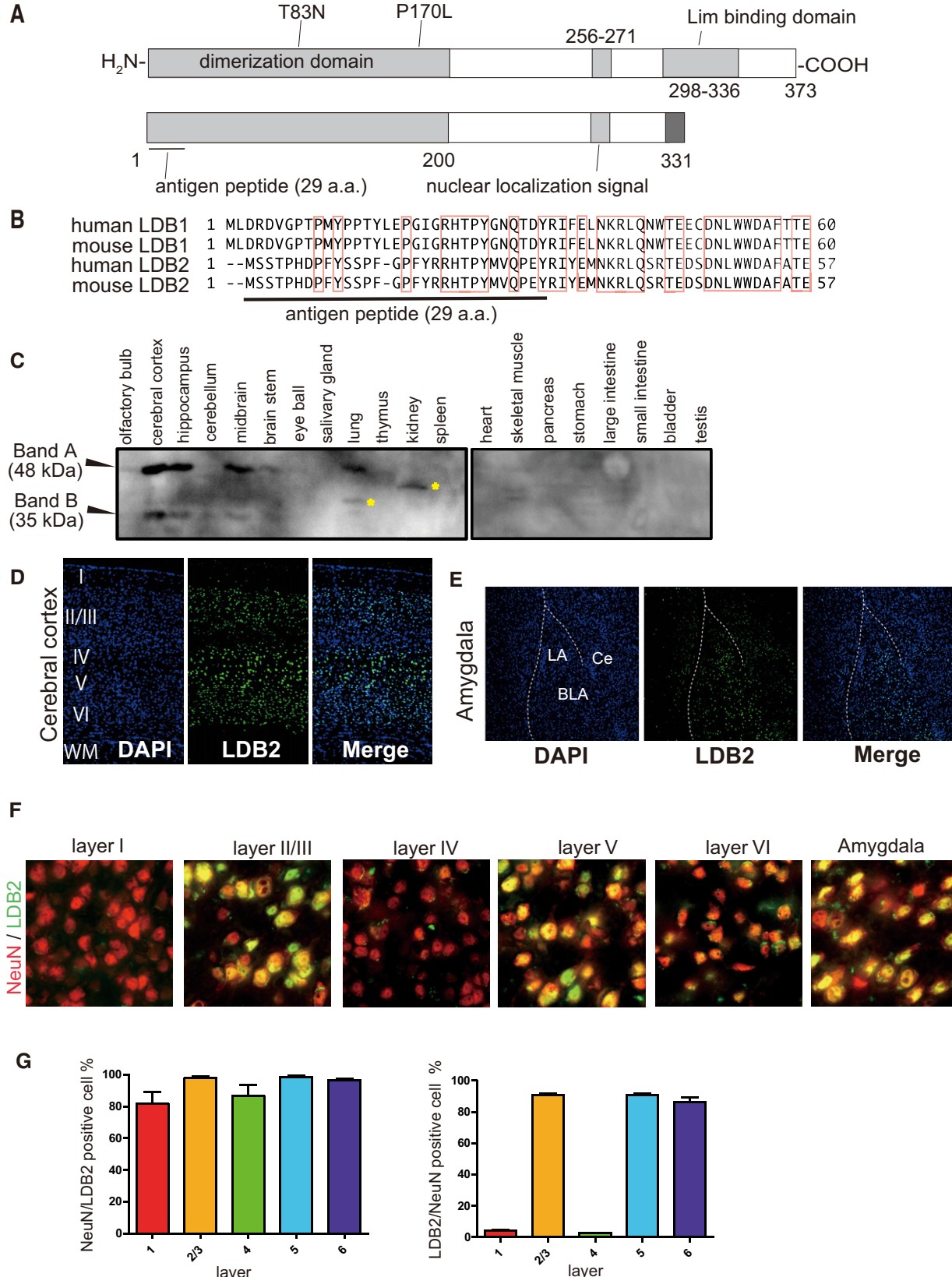

**Figure 1.**

the *Ldb2* knockout (KO) mice, indicating that these two bands correspond to the *Ldb2* gene products (see the next section). The expression of LDB2 in the brain was evident in the frontal cortex, hippocampus, and midbrain, but not in the cerebellum or brain stem (Fig 1C). While the 48-kDa band was also seen in the lung, the other tissues examined showed no discrete band. Immunohistochemical analyses using a rat monoclonal anti-Ldb2 antibody (clone 4-1F) revealed that LDB2 is expressed exclusively in the nuclei of neurons in the brain (Fig 1D–G). In the cerebral cortex, > 80% of LDB2-positive cells were NeuN-positive in the layers II/III, V, and VI, showing neuron-selective expression of LDB2 (Fig 1D, F, and G). Here, it is of note that most of Neu-positive cells in the layers I and IV were LDB2-negative. In the hippocampus, expression appeared to be restricted in the pyramidal cell layer in the adult stage (Fig EV1F). In addition, a selective expression was seen in neurons in the amygdala (Fig 1E, right panel). Consistent with the Western blot data, signals were rarely seen in the olfactory bulb, striatum, thalamus, or cerebellum (Fig EV1F), suggesting the region-specific roles of LDB2 in the brain. We showed most of the LDB2 signals disappeared in the cerebral cortex, hippocampus, and amygdala from *Ldb2* KO mice (Fig EV1A–F), evidencing specificity of the antibody used (Fig EV1F).

### *Ldb2* KO mice display pleiotropic behavioral abnormalities relevant to mental disorders

We next set out to analyze phenotypes of *Ldb2*-deficient mice (Fig 2, Fig EV1, Appendix Table S1) employing our behavioral test batteries (Watanabe *et al*, 2007; Ohnishi *et al*, 2014, 2019; Shimamoto-Mitsuyama *et al*, 2020). In order to minimize the effects of the complex (mixed) genetic background, the mice used were backcrossed for > 6 generations to the inbred C57BL/6N (B6N) mice before intercrossing between heterozygotes to create homozygotes. In homozygotes, the *Ldb2* transcript was undetectable without any compensatory upregulation of the *Ldb1* transcript (Fig EV1B). Moreover, homozygotes lacked both the 48 and 35 kDa bands (Fig EV1C), confirming that these two bands correspond to the *Ldb2* gene products. In contrast to the *Ldb1* KO mice, which were reported to die during the gastrulation with severe patterning defects (Mukhopadhyay *et al*, 2003), *Ldb2* KO mice grew normally without any morphological abnormalities in their body or brain (Fig EV1D and E).

The *Ldb2* KO mice exhibited hyperactivity in the home cage and open-field test (Fig 2A and B), the phenotype seen in multiple pharmacological and genetic animal models for schizophrenia or the manic state of bipolar disorder (Enomoto *et al*, 2007). Acoustic startle response (Fig 2C) and prepulse inhibition (Fig 2D) of the KO mice were comparable with those of the WT littermates. In the Morris water-maze test, the KO animals were not hyperactive, but showed longer no-movement time and slower swimming speed

when compared to the wild-type (WT) littermates, resulting in a longer latency-to-escape (Fig 2E), which suggests impaired ability of the KO animals to sustain motivation. On the other hand, the KO mice exhibited no abnormality in path length in the hidden test or stay time in the target quadrant in the probe test, collectively indicating no clear deficits in spatial memory of the KO animals. In the Y-maze test, the KO mice exhibited a significant reduction in the spontaneous alternation (Fig 2F), suggesting an impairment in working memory. Notably, the KO mice exhibited shorter freezing time than the WT littermates in both contextual (a trend, $P = 0.054$) and cued ($P = 0.0003$) tests in the fear-conditioning paradigm where electrical foot shock and tone were presented as an unconditioned stimulus (US) and a conditioned stimulus (CS), respectively (Fig 2G). These results are not likely due to impaired hearing ability or nociception in the KO animals (Fig 2C and H), but suggest severe deficits in attention and/or coupling US with CS and abnormalities in neuronal circuits for fear conditioning in the KO mice.

Next, we examined a sensitivity of the *Ldb2* KO mice to two psychoactive drugs. In the open-field test, the KO mice were more sensitive to methylphenidate (3 mg/kg, s.c.), a dopaminergic psychostimulant, than the WT littermates (Fig 3A). Similarly, the KO mice were more sensitive to a non-competitive *N*-methyl-ᴅ-aspartate (NMDA) receptor blocker, MK-801 (0.2 mg/kg, s.c.) known as a hallucinogenic drug (Fig 3B). In addition, MK-801 (0.5 mg/kg) injection evoked jumping (or popping) behavior when the cylinder was mildly tilted (Fig 3C), supporting hypersensitivity to the hallucinogenic drug and suggesting an increased impulsivity of the *Ldb2* KO mice. Importantly, deficits in fear-conditioning test of the KO mice were ameliorated by the treatment with an atypical antipsychotic (clozapine) (Fig 4A). Moreover, the hyperlocomotive trait of the KO animal in the home cage was alleviated by chronically feeding a chow containing 0.24% (w/w) lithium carbonate, the first-line mood-stabilizing agent for treating bipolar disorder (Fig 4B). Collectively, the *Ldb2* KO mice at least partly fulfill the criteria of both face validity (a similarity exists between behavioral phenotypes of a model animal and disease symptoms) and predictive validity (a drug used for a disease is also effective to ameliorate phenotypes of a model animal) for schizophrenia and bipolar mania.

### Amygdala of *Ldb2* KO mice displays aberrant neuronal response to unconditioned/conditioned stimuli

The fear learning of *Ldb2* KO mice was severely impaired especially in the cued paradigm. It is well known that the amygdala, where significant expression of LDB2 was seen (Fig 1E), plays an indispensable role in the cued paradigm. Therefore, we examined whether the KO mice show any biological deficits in the amygdala, by visualizing neuronal activity using transgenic mice expressing the fluorescent protein Venus under the control of the *Arc/Arg3.1*

**Figure 2. Behavioral characterization of the *Ldb2* KO mice.**

A–H   *Ldb2* KO mice (red, *n* = 10–11) were compared to control (blue, *n* = 10–11) in various behavioral tests. Home cage-activity test (A). Voluntary locomotor activities in the home cages were monitored for a week. Data of dark (left) and light (right) phases are indicated. Open-field test (B), acoustic startle response (C), prepulse inhibition test (D), Morris water-maze test (E) where data in hidden test (escape latency and moving speed, path length and no-movement time) and probe test (path length) are shown, and Y-maze test (F). Fear-conditioning test (G). Results of contextual (left) and cued (right) tests are shown. Cumulative values are shown at the bottom. Hot-plate test (H). *$P$ < 0.05, **$P$ < 0.01, ***$P$ < 0.001 [Tukey test except for F and G (bottom) where *t*-test was used]. The error bars represent SEM.

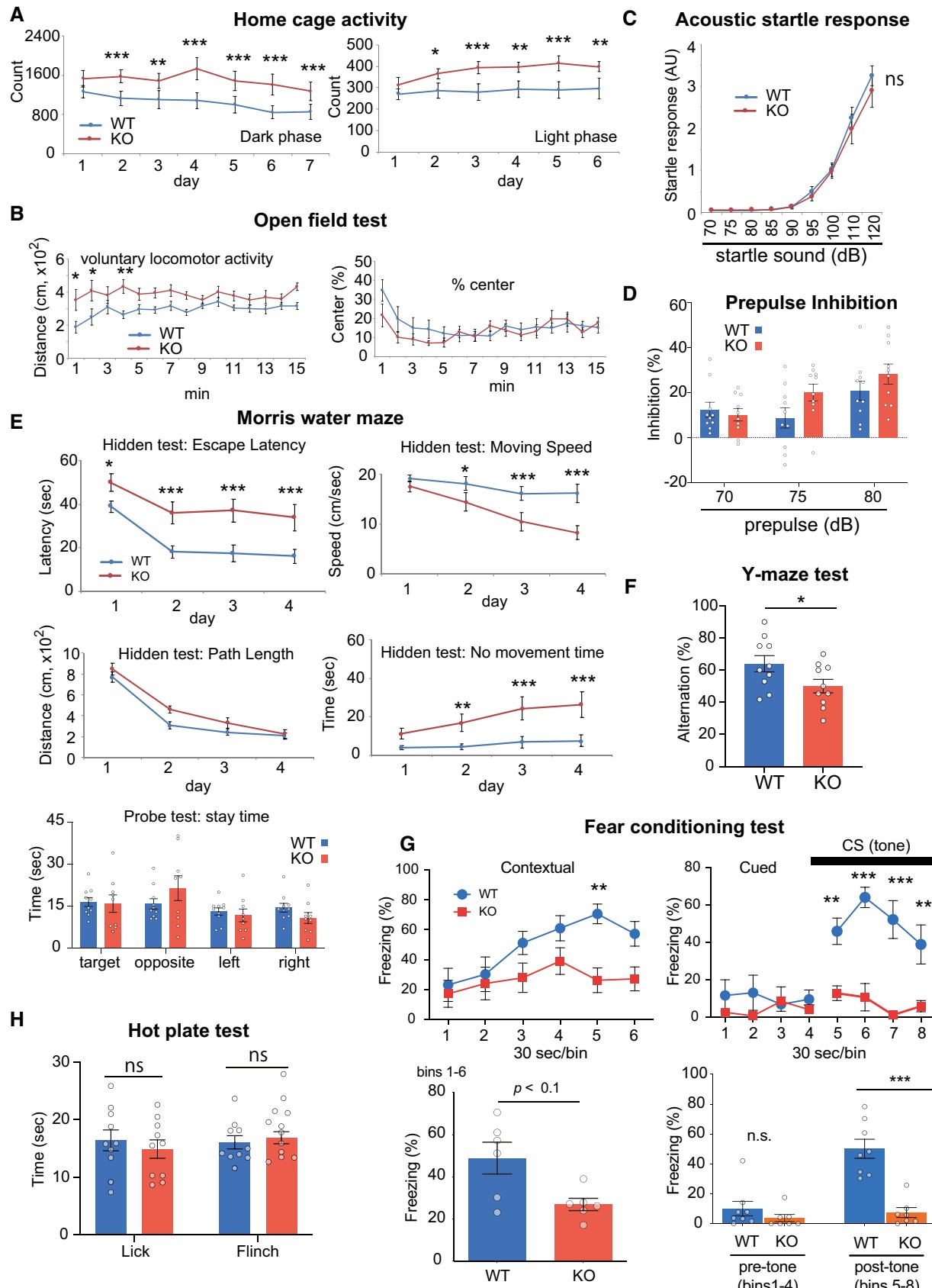

**Figure 2.**

promoter (Arc-Venus) (Mikuni *et al*, 2013; Vousden *et al*, 2015). Here, *Arc* is one of the immediate early genes and is known to be transcribed in an activity-dependent manner (Miwa *et al*, 2008). Importantly, the time resolution is expected to be significantly high, since the fluorescent protein Venus is fused to the PEST sequence, which destabilizes the protein of interest by rapid degradation. We confirmed that Venus-positive neurons certainly exist in the basolateral complex of the amygdala (Appendix Fig S3A), and most Venus-positive neurons also express endogenous ARC protein (Appendix Fig S3B). We generated Arc-Venus tg/tg; *Ldb2* KO (AV-KO), and Arc-Venus tg/tg; *Ldb2* WT (AV-WT) mice to monitor neuronal activities in the amygdaloid complex through the Venus fluorescence.

Because the lack of LDB2 caused severe deficits only in the cued-fear learning that is known to depend on synaptic plasticity in the lateral amygdala (LA), LDB2 may play an important role in synaptic transmission especially in the LA of the amygdala. To address this possibility, we next conducted visualized patch-clamp experiments in the LA of the tone-shock pairing-experienced AV-WT and AV-KO mice. Miniature excitatory postsynaptic currents (mEPSCs) recorded in the LA of Venus-positive and -negative neurons of the LA in AV-WT and AV-KO mice were similar among the groups (median amplitude: AV-WT/Venus$^+$, 9.98 ± 0.76 pA; AV-WT/Venus$^-$, 11.32 ± 0.98 pA; AV-KO/Venus$^+$, 10.11 ± 0.71 pA; AV-KO/Venus$^-$, 11.60 ± 0.94 pA; *Ldb2* genotype, $P = 0.889$; Venus expression, $P = 0.152$; interaction, $P = 0.872$; mean frequency: AV-WT/Venus$^+$, 13.66 ± 3.35 events/s; AV-WT/Venus$^-$, 16.76 ± 2.75 events/s; AV-KO/Venus$^+$, 15.50 ± 2.75 events/s; AV-KO/Venus$^-$, 14.97 ± 2.13 events/s; *Ldb2* genotype, $P = 0.992$; Venus expression, $P = 0.644$; interaction, $P = 0.513$ by two-way ANOVA, $n = 11, 13, 11, 15$, respectively; Fig EV2A and B). For all the recorded Venus-positive neurons, the fluorescence ratio of their cell-body area over their background area, which represents their relative fluorescence intensities, was not significantly different between AV-WT and AV-KO mice (Fig EV2C). Interestingly, the ratio of the fluorescent intensity of recorded Venus-positive cells over their median amplitude of the mEPSCs showed significant inverse correlation ($r = -0.515$, $n = 22$, $P = 0.014$; Fig 5A), and the ratio of the fluorescent intensity over the mean frequency of mEPSCs did as well ($r = -0.528$, $n = 22$, $P = 0.012$; Fig 5B). These results indicate that synaptic activity is more reduced when neurons express more ARC and that the expression of the α-amino-3-hydroxy-5-methyl-4-isoxazolepropionic acid receptor (AMPAR) in the postsynaptic spine is inversely correlated with the expression of ARC, although there was no difference in mEPSC parameters between genotypes.

Since synaptic activity might influence the amount of *Arc* gene expression, we next analyzed the effect of fear conditioning on the amount of Arc expression in the amygdala. Mice were divided into three groups depending on their conditioning trials. The "Paired"

group received 8 tone-shock pairings during the fear-conditioning trial. The "Unpaired" group received 8 tones and shocks separately. The "Naïve" group was kept in a home cage without fear conditioning (Fig EV3A). All mice were deeply anesthetized and perfused with 4% formaldehyde 3 h after the conditioning stimuli (except for Naïve group; Fig EV3A), and then, their brains were removed to prepare brain slices containing the amygdaloid complex. We analyzed three coronal sections of each of the left and right amygdala in each mouse. Because Venus-positive neurons were particularly conspicuous in the lateral amygdala (LA) and the lateral side of the basal amygdala (BAl) in the Paired mice (Fig EV3B, see also Fig 6A), we focused on these areas. In the LA (Fig 6B, upper panel), the number of Venus-positive neurons of AV-WT mice was significantly increased only in the Paired group compared to the Naïve and Unpaired groups, indicating that the tone-shock pairing activated LA neurons. On the other hand, the number of Venus-positive neurons of AV-KO mice was significantly increased in the Unpaired as well as the Paired groups compared to the Naïve group. Furthermore, the numbers of Venus-positive neurons both in the Unpaired and Paired AV-KO mice were also significantly increased compared to the same groups of the AV-WT mice. In the BAl (Fig 6B, lower panel), the numbers of Venus-positive neurons were significantly increased both in the Unpaired and Paired groups compared to the Naïve group in both the AV-WT and AV-KO mice. However, significant differences between genotypes were not seen in any treatment groups. To examine what kind of neurons was activated by the paired fear-conditioning trial, we analyzed Venus-positive neurons in the LA by immunohistochemistry with antibodies against neuronal markers. We found that most Venus-positive neurons were CaMKIIα-positive in both the AV-WT (97.1 ± 1.8%) and AV-KO mice (93.5 ± 2.5%) (Fig EV3C and D). Likewise, most Venus-positive neurons were GAD67 negative in both the AV-WT (98.6 ± 1.1%) and AV-KO mice (97.5 ± 1.5%) (Fig EV3C and E). These results indicate that the lack of LDB2 expression in excitatory neurons in the LA of the amygdala promoted the expression of ARC in response to foot shocks during both paired and unpaired fear-conditioning trials.

Since the Venus fluorescence intensity reflects the amount of *Arc* gene transcription, we reanalyzed relative fluorescence intensities of Venus-positive neurons in the LA. According to fluorescence intensities, we divided Venus-positive neurons into three categories: Weak [from 15 (threshold) to 27 arbitrary units (AU)], Middle (from 28 to 48 AU), and Strong (more than 49 AU) so that Venus-positive neurons of the Naïve group of AV-WT mice are divided into three equal parts. In the AV-WT mice, only Weak Venus-positive neurons were significantly increased in the Paired group compared to the Naïve group (an asterisk in Fig 6C, left panel), indicating that *Arc* gene expression in response to the Paired stimuli is relatively weak. In the AV-KO mice, both Weak and Middle Venus-positive neurons were increased in the Paired group compared to the Naïve group

**Figure 3. Increased sensitivity to hallucinogenic drugs in the *Ldb2* KO mice.**

A, B Locomotor activity was assessed in the same experimental set as in the open-field test. After acclimation for 120 min, mice ($n = 10$–12/group) were injected (arrows) with methylphenidate (MPD: 3 mg/kg, A) or MK-801 (0.2 mg/kg, B) and their locomotor activity was monitored. In the right panels, cumulative values for pre- and postinjection times are also shown. All data are shown with error bars representing SEM. *$P < 0.05$, **$P < 0.01$, ***$P < 0.001$ (Tukey test).

C Popping behavior induced by the injection of MK-801 (0.5 mg/kg) was manually assessed in WT ($n = 4$) and KO ($n = 6$) animals. Typically, each episode is constituted by blocks of 5–15 successive jumping behaviors. Episode numbers in 20 min (left) and total duration (right) are shown in a box-whisker plot. *$P < 0.05$, **$P < 0.01$ (Mann–Whitney *U*-test). Data are shown with error bars representing SEM.

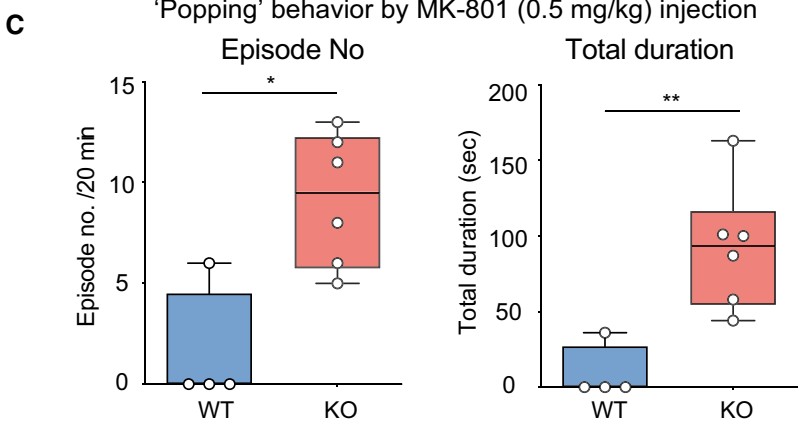

**Figure 3.**

(asterisks in Fig 6C, right panel). Especially, the increment of the Middle Venus-positive neurons was significantly larger in the AV-KO compared to AV-WT (hash marks in Fig 6C, right panel). These results indicate that the lack of expression of LDB2 promotes the *Arc* gene transcription responding to the tone-shock paired stimuli in the LA neurons.

Collectively, it is strongly suggested that (i) the LDB2 protein in LA neurons suppresses *Arc* gene expression during fear conditioning, (ii) the amount of cellular *Arc* expression is inversely correlated with spontaneous synaptic activity in the LA, and (iii) the *Ldb2* KO mice display *Arc* overexpression especially in the LA. Thus, *Arc* overexpression in the LA might lead to the impairment in tone-related fear conditioning.

## Neurogenesis is reduced in the hippocampus of *Ldb2* KO mice

*Ldb2* KO mice displayed a tendency to impaired freezing in the contextual paradigm of the fear-conditioning test (Fig 2G). Neurogenesis in the hippocampus has a role in this learning (Whoolery *et al*, 2017; Huckleberry *et al*, 2018). In addition, impairments in adult neurogenesis have been repeatedly reported in schizophrenia, affective disorders, and their model animals (Gonçalves *et al*, 2016; Apple *et al*, 2017). We found that the neurogenesis in hippocampal dentate gyrus (DG) was remarkably decreased in the KO mice (Appendix Fig S4).

## Ldb2 forms complexes with known transcription-related proteins

To obtain insights into how LDB2 exerts multiple biological functions, we set out to screen for binding partners for LDB2 utilizing the yeast two-hybrid system and the cDNA library from the human embryonic brain. We identified several candidates for the binding partners: single strand binding protein (SSBP) family members (SSBP2, 3, and 4), Lim domain-only (LMO) family members (LMO2, 3, and 4), and Lim home-obox (LHX) family member (LHX2; Table 1), most of which have been reported to be expressed in the brain and regulate gene expression. Other candidates, including actinin α1/2 (ACTN1/2), translin-associated factor X (TSNAX), pyruvate kinase, muscle-type (PKM2), and CD9 and/or sulfide quinone oxidoreductase (SQOR) (Ide *et al*, 2019), were also identified (Table 1). Since two-hybrid screening is known to sometimes produce false-positive signals, we validated the specificities of the potential interactions with co-immunoprecipitation assay by focusing on the proteins that potentially act as transcription factors or transcription regulatory factors. The N-terminal region of LDB2 is known to mediate homodimerization (Matthews & Visvader, 2003) (Fig 1A). The LDB2-LDB2 interaction was detectable in our assay (Fig 7A). Also, the interaction between LDB2 and each of LHX2, LMO2, LMO3, LMO4, SSBP2, SSBP3, and SSBP4 was confirmed (Fig 7B–D).

The interaction between LDB1/2 and RLIM (also known as RNF12) is reported (Bach *et al*, 1999; Ostendorff *et al*, 2002; Hiratani *et al*, 2003; Xu *et al*, 2007). RLIM serves as a ubiquitin ligase E3 that degrades multiple proteins, including LDB1/2 *via* the proteasome pathway. Kataoka *et al* (2016) reported the *de novo* missense variant *RLIM* S455F in a bipolar proband by exome sequencing. We detected the physical interaction between LDB2 and RLIM, but it was not affected by the *RLIM* variant (Fig 7E). Horiuchi *et al* (2020) identified

two rare missense variants (T83N and P170L) (Fig 1A) in bipolar patients by resequencing of *LDB2*. The LDB2 P170L mutant seemed to enhance or stabilize the interaction with RLIM (Fig 7E, left panel) with unknown mechanism. Interestingly, the LDB2 T83N and LDB2 P170L mutants were more abundant in the input fraction than the wild-type protein in some experiments (Fig 7A, B, C, and E), leading to the increased LDB2 protein co-immunoprecipitated with its binding partners. In the lysate of adult mouse brain, SSBP3 and LMO3 were co-purified with LDB2 (Fig 7F), corroborating that the LDB2 protein forms transcription complexes with them in the brain.

## ChIP-seq analysis

It is still unclear how the LDB2 protein controls gene expression in the brain. To gain insights into this mechanism and expand our findings to humans, ChIP-seq analysis was performed using an LDB2 specific antibody and human-induced pluripotent stem (iPS) cells-derived neurospheres (Fig 8A). The neurospheres are constituted by neural stem cell-like populations (Toyoshima *et al*, 2016).

The motif analysis with MEME-ChIP (http://meme-suite.org/tools/meme-chip; Machanick & Bailey, 2011) revealed that the consensus binding sequences for the EGR (early growth response) transcription factors were significantly enriched in the central segments of the ChIP-seq-mapped regions (Fig 8B), raising the possibility that LDB2 indirectly binds to target DNA regions *via* the EGR family members. The EGR family members (EGR1-4), immediate early gene products, are well-known transcription factors that function in the calcineurin signal transduction pathway (Yamada *et al*, 2007). We previously reported the genetic association of *EGR3* with schizophrenia and downregulation of the transcripts for *EGR1*, *EGR2*, and *EGR3* in the postmortem brain samples from patients with schizophrenia (Yamada *et al*, 2007). Indeed, ChIP-seq signal for the EGR1 protein in cultured cell lines (ECC1: endometrial cancer; K562: erythromyeloblastoid leukemia) overlapped that of the LDB2 protein in neurosphere cells at the promoter region of *ARC* (Fig 8C). Genome-wide analysis revealed that more than 3,829 genes identified in the LDB2-ChIP-seq analysis, including *ARC*, were found to harbor the EGR1-ChIP-seq signals in ECC1 cells (Fig 8D and E). A substantial fraction of the peaks was overlapped with those obtained with EGR1-ChIP-seq in three other cell lines H1ES (an ES cell line), HCT116 (a colon cancer cell line), and K562 (Appendix Table S2, Fig EV5 A and B), further indicating interaction between LDB2 and EGR1 proteins. Importantly, about 50% of the common ChIP peaks between LDB2 and EGR1 in the ECC1 and K562 cell lines were enriched in the upstream regions or 5' UTRs of target genes (Fig 8E, Appendix Table S3). A Gene Ontology analysis (http://geneontology.org; Ashburner *et al*, 2000; Mi *et al*, 2019) using the set of genes (Appendix Table S3) underscored that the biological process "Regulation of dendrite spine morphogenesis" exhibits the second lowest FDR (false discovery rate) with a fold

---

**Figure 4. Therapeutic drugs ameliorate behavioral phenotypes of KO mice.**

A   Fear-conditioning test. Freezing (%) was monitored in the cued (top) and contextual (bottom) tests before (pre) and after (post) administration of clozapine (left) or haloperidol (right). Data are presented as means ± SEM (*n* = 7–9/group). *P < 0.05; ***P < 0.001 (Tukey test); #P < 0.05 (*t*-test, KO/saline vs. KO/clozapine).

B   Home cage-activity test. Spontaneous locomotor activity of mice chronically fed chow with or without 0.24% (w/w) lithium carbonate for two months was monitored for 6 days in the home cage. Data are presented as means ± SEM (*n* = 9–13/group). *P < 0.05 (Tukey test).

**A**

# Fear conditioning test

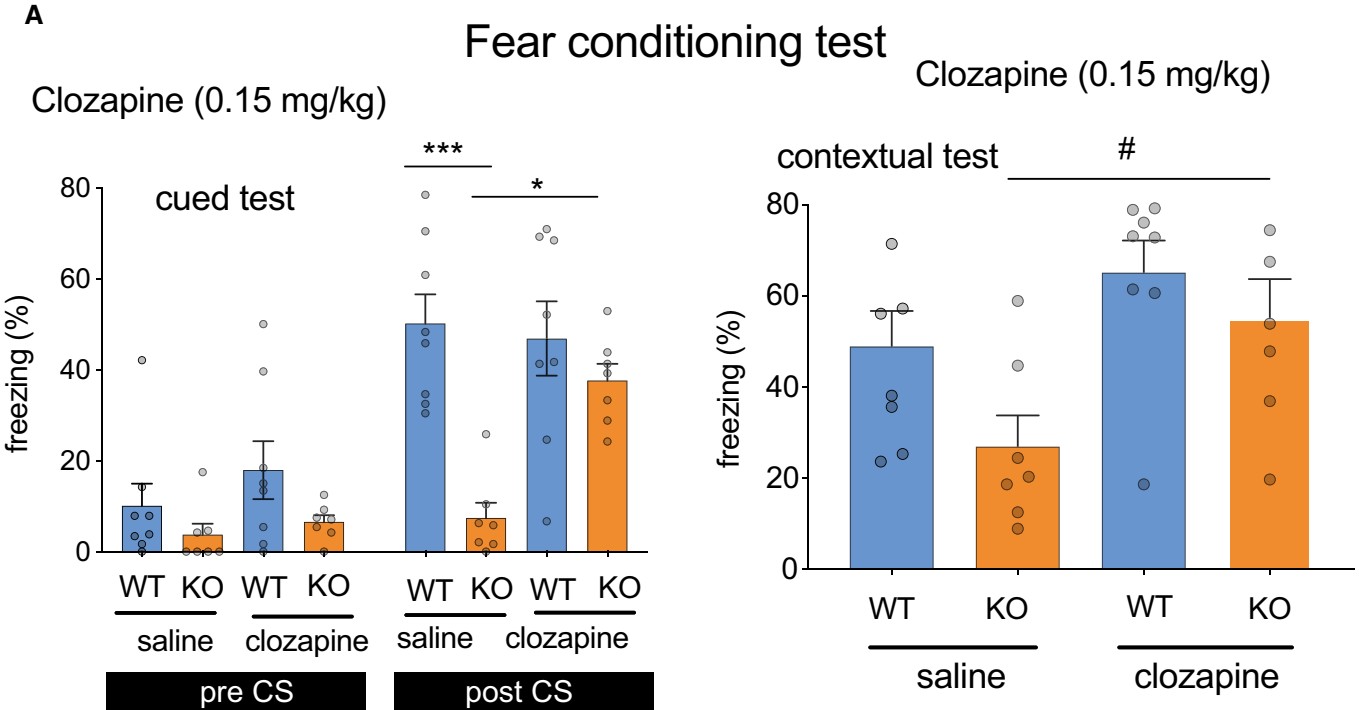

**B**

# Home cage activity test

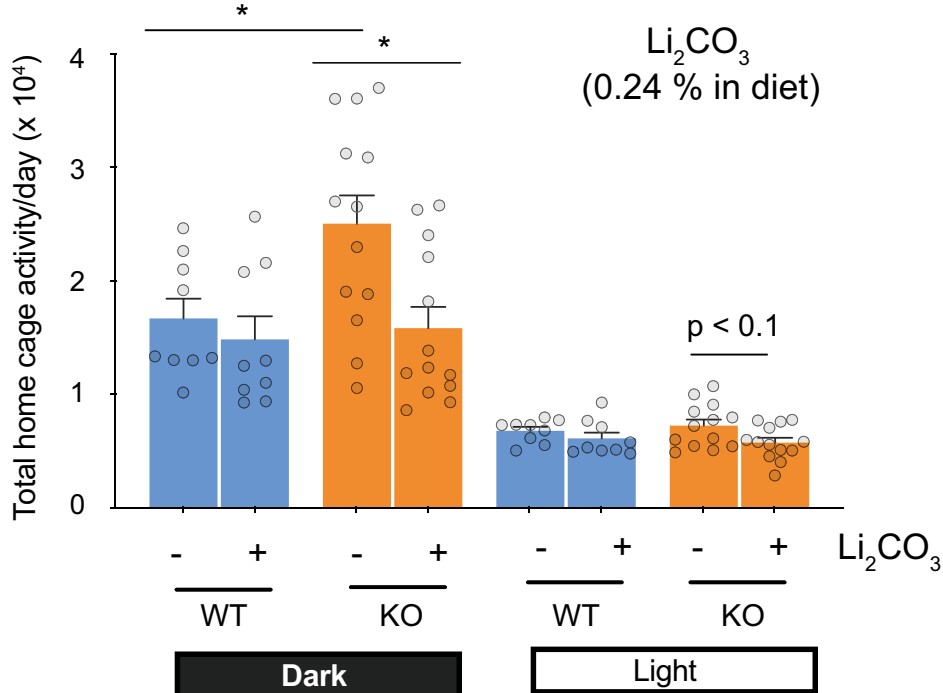

**Figure 4.**

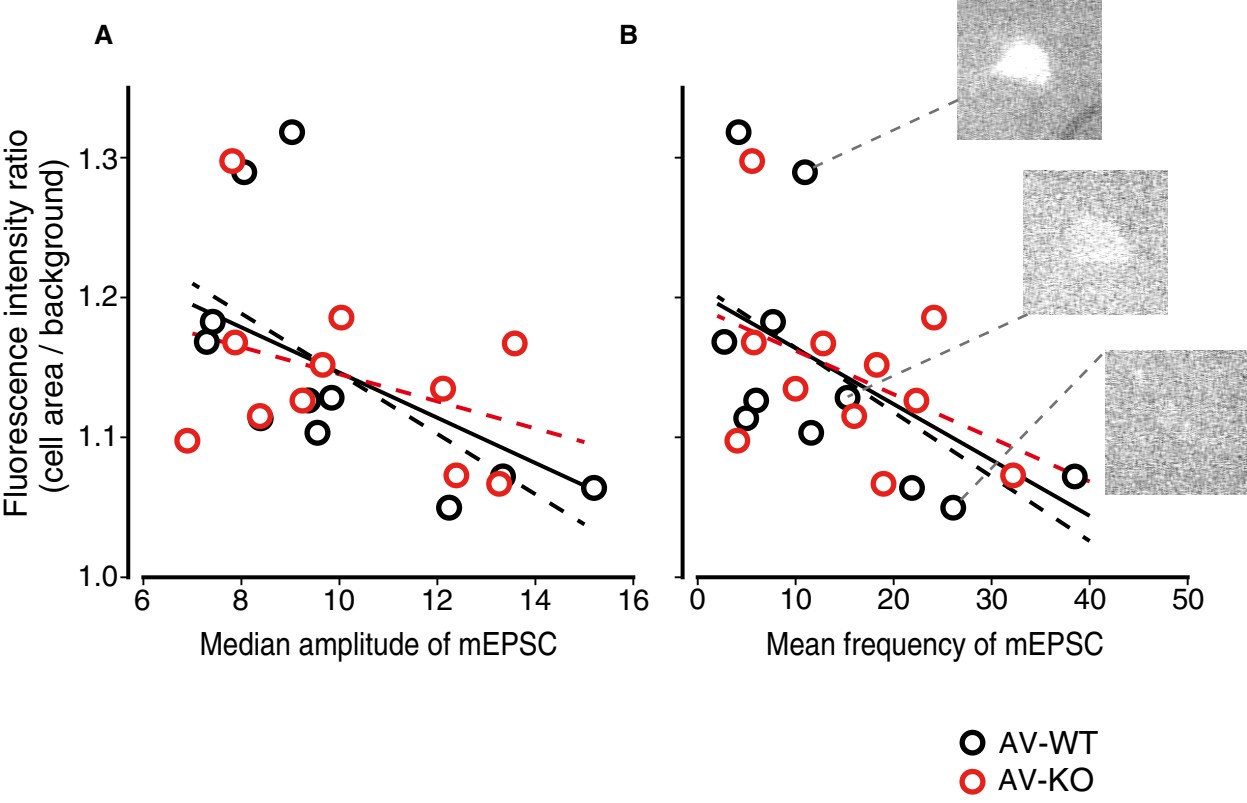

**Figure 5.** **The amplitude and frequency of miniature EPSCs of Venus-positive cells are inversely correlated with the fluorescence intensity.**

A   Scatter plots showing Venus-fluorescent intensities and the median amplitude of miniature EPSCs (mEPSC) in the LA of the AV-KO mice and their littermates (AV-WT).
B   Scatter plots showing Venus-fluorescent intensities and the mean frequency of miniature EPSCs in the LA of the AV-KO and their control littermates (AV-WT). Three typical pictures of Venus-positive cells with various fluorescent intensities are shown. Lines in each panel are regression lines showing inverse correlation of the fluorescent intensity with the mEPSC amplitude (A) or the mEPSC frequency (B). The black dotted lines are regression lines for AV-WT mice, the red dotted lines are those for AV-KO mice, and black solid lines are those for all cells.

enrichment of 3.2 (Table 2), in which process *ARC* was included (Appendix Table S4).

As our *in vitro* binding assay did not detect a direct interaction between LDB2 and EGRs, we speculate that LDB2 cooperates with the EGR family via unknown protein(s) to control expressions of numerous genes including *ARC* (Fig 9) and that the "LDB2-EGR axis" may potentially underlie the etiology of subset of mental disorders.

## Discussion

In the current study, we delved into a specific schizophrenia case with a balanced chromosomal translocation t(4;13)(p16.1;q21.31). The *LDB2* gene is mapped to the 5'-upstream region of the breakpoint within chromosome 4 (Appendix Fig S1). *LDB2* and the flanking genomic regions are located on the same topologically

**Figure 6.** **The number of Venus-positive neurons in the lateral amygdala is larger in fear-conditioned KO mice.**

A   A schematic image of the basolateral complex of the amygdala.
B   Numbers of Venus-positive neurons in the lateral nucleus (LA) and the lateral side of the basal nucleus (BAl) of Naive, Unpaired, and Paired groups. Open and filled bars represent AV-WT and AV-KO mice, respectively. Positive neurons of AV-WT was increased only in the Paired group compared to the Naïve and Unpaired groups (*F* [2,14] = 4.26, *P* = 0.040; after Fisher's LSD, *P* = 0.022 compared to Naïve, *P* = 0.032 compared to Unpaired) in the LA (upper panel). AV-KO was increased in the Unpaired as well as the Paired groups compared to the Naïve group (*F*[2,14] = 8.66, *P* = 0.005; after Fisher's LSD, Unpaired, *P* = 0.003; Paired, *P* = 0.006). Positive neurons both in the Unpaired and Paired AV-KO mice were increased compared to the same groups of the AV-WT mice (Unpaired, *P* = 0.014; Paired, *P* = 0.006 by Student's *t*-test). In the BAl (lower panel), positive neurons were increased both in the Unpaired and Paired groups compared to the Naïve group in both the AV-WT (*F* [2,14] = 10.7, *P* = 0.002; after Fisher's LSD, Unpaired, *P* = 0.013; Paired, *P* = 0.001) and AV-KO mice (*F*[2,14] = 8.61, *P* = 0.003; after Fisher's LSD, Unpaired, *P* = 0.013; Paired, *P* = 0.005). The error bars represent SEM (*n* = 5/group).
C   Numbers of Venus-positive neurons classified by fluorescence intensities in the LA of the AV-WT and AV-KO mice. Weak positive neurons were increased in the Paired group compared to the Naïve group in AV-WT (*P* = 0.017, Student's *t*-test, *n* = 5, respectively; an asterisk in left panel). Both Weak and Middle positive neurons were increased in the Paired group compared to the Naïve group in AV-KO (Weak, *P* = 0.015; Middle, *P* = 0.003; Student's *t*-test, *n* = 5, respectively; asterisks in right panel). The increment of the Middle Venus-positive neurons was larger in the AV-KO compared to AV-WT (genotype × group interaction: *F*[1,16] = 10.2, *P* = 0.006, two-way ANOVA; a hash marks in right panel). The error bars represent SEM (*n* = 5, respectively). **P* < 0.05, ***P* < 0.01 (compare to Naïve, *t*-test), ##*P* < 0.01 (compared to WT, *t*-test).

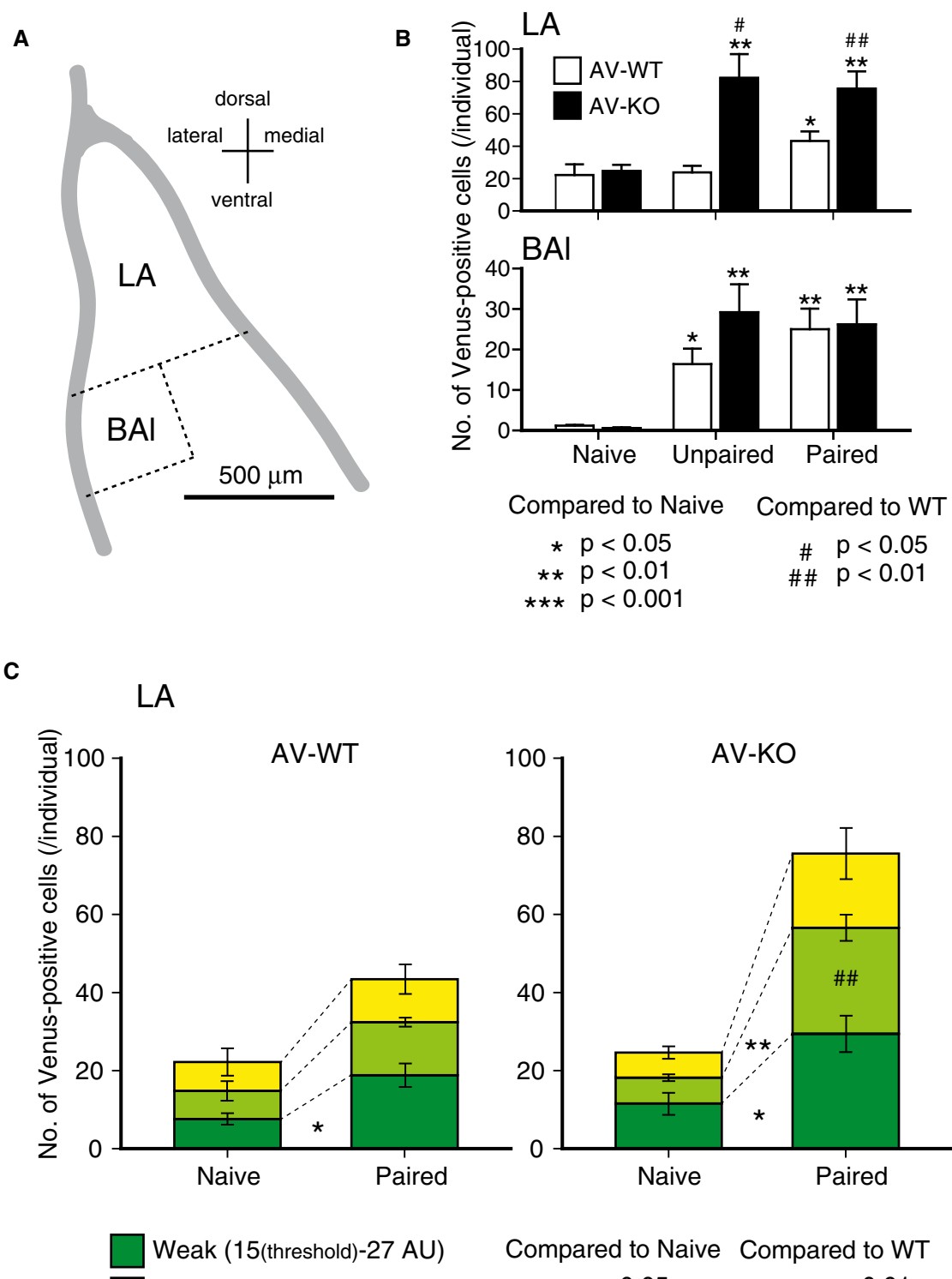

**Figure 6.**

associating domains (TAD), and the breakpoint overlaps with DNase I hypersensitive (DHS) sites and long terminal repeats (LTRs), and the flanking regions harbor signals for enhancers/ enhancer-lncRNAs. Conformally, we detected a lower expression of *LDB2* in iPS cells and iPS cell-derived neurospheres from the patient compared to a healthy control (Appendix Fig S2). In

**Table 1. Summary of yeast two-hybrid screening for LDB2 binding protein**

| Gene | Clone number | Annotation |
|---|---|---|
| SSBP2 | 25 | single-stranded DNA-binding protein 2 |
| SSBP3 | 11 | single-stranded DNA-binding protein 3 |
| SSBP4 | 26 | single-stranded DNA-binding protein 4 |
| LMO2 | 13 | LIM domain only 2 |
| LMO3 | 4 | LIM domain only 3 |
| LMO4 | 28 | LIM domain only 4 |
| LHX2 | 2 | LIM homeobox 2 |
| ACTN1 | 1 | actinin, alpha 1 |
| ACTN2 | 2 | actinin, alpha 2 |
| TSNAX | 1 | translin-associated factor X |
| NECAB2 | 1 | N-terminal EF-hand calcium binding protein 2 |
| SQRDL | 1[a] | sulfide quinone reductase-like |
| CD9 | 1[b] | CD9 molecule |
| PKM2 | 1 | pyruvate kinase, muscle |
| UBE4A | 1[c] | ubiquitination factor E4A |
| CCDC39 | 1[d] | coiled-coil domain containing 39 |
| sum | 119 | |

[a]This clone contained the CD9 sequence, too.
[b]This clone contained the SQRDL sequence, too.
[c]This clone contained in-frame SSBP4 and out-of-frame UBE4A sequences.
[d]This clone contained in-frame SSBP2 and out-of-frame CCDC39 sequences.

the future study, the cellular phenotypes of these cells should be addressed.

Multiple observations in the current study favor to the idea that LDB and LDB2-mediated gene expression networks play a potential role in the pathogenesis of schizophrenia and other mental disorders, including bipolar disorder. First, mice express Ldb2 in neurons at the cerebral cortex, hippocampus, and amygdala, all of which are crucial anatomical regions for higher brain function. Second, ablation of Ldb2 in a mouse resulted in pleiotropic behavioral abnormalities reminiscent of schizophrenia and bipolar mania. Those phenotypes include hyperactivity in home cage or open-field, reduced alternation in the Y-maze test, severe deficits in fear conditioning, and exaggerated responses to methylphenidate or MK-801. Similar phenotypes have been repeatedly reported in many of animal models for schizophrenia or bipolar disorder including Disc1 gene modified mice (Clapcote et al, 2007). Moreover, some

of behavioral alternations were alleviated by an antipsychotic (clozapine) or a mood stabilizer (lithium), suggesting that the Ldb2 KO animal satisfies the predictive validity, which is justified by expected responses of the model to therapeutics including medication. The animal also satisfies the face validity (a similarity exists between behavioral phenotypes of a model animal and disease symptoms) and construct validity (a model animal can be generated based on a mechanistic theory of disease; in the present study, LDB2 locus is disrupted in a schizophrenia case and the mouse homolog was knocked out). However, it should be noted that prior GWAS has not supported that common variants in the LDB2 locus contribute to the genetic susceptibility of schizophrenia (Schizophrenia Working Group of the Psychiatric Genomics Consortium, 2014) or bipolar disorder (Stahl et al, 2019).

We identified SSBP, LHX, and LMO family members, all of which are well-characterized gene expression regulators, as direct interactors of LDB2. It is of interest that heterozygous KO mice for Lmo4, which encodes one of LDB2 interactors identified in this study, exhibited enhanced freezing in the fear-conditioning test (Maiya et al, 2012). LMO4 is expressed in the pyramidal neurons in the amygdala in mice (Maiya et al, 2012). It is tempting to hypothesize that LMO4, which also lacks DNA-binding domain and is believed to interact with the LDB family via its LIM domain, antagonizes the function of LDB2, affecting the control of the freezing behavior in the amygdala.

Our ChIP-seq analyses suggested that binding sites of the LDB2 and EGR proteins in human cells were concentrated to synaptogenesis-related genes including ARC. The expression of Arc was modulated in the amygdala of Ldb2 KO mice. Therefore, it is tempting to speculate that synaptogenesis is one of the downstream targets for LDB2 in the amygdala, which probably underlies the deficit in attention and/or fear conditioning of the Ldb2 KO mice. Our study revealed that the amount of cellular Arc expression is negatively correlated with synaptic activity in the LA and that Ldb2 deficiency results in an excess of the Arc expression only in the LA, which probably causes the disturbance of auditory fear learning. It is well established that ARC controls synaptic plasticity through the postsynaptic mechanism. In the dendritic spines, accumulation of deSUMOlyated ARC leads to downregulation of AMPARs and destabilization of F-actin and eventually induces spine shrinkage (Newpher et al, 2018). Disturbance of Arc/ARC expression has been reported in some neuropsychiatric diseases. In Fragile X syndrome (FXS), ARC expression is enhanced by a reduction of the FMRP protein, a repressor of translation. In FXS and its model animals, immature spine morphology is commonly observed, most likely due to the overexpression of ARC/Arc (Newpher et al, 2018).

**Figure 7. LDB2/Ldb2 forms complexes with known transcription regulators.**

A–E Results of co-immunoprecipitation between overexpressed LDB2 and interaction candidates are shown. LDB2 vs. LDB2 (A), LDB2 vs. LHX2 (B), LDB2 vs. LMO2/3/4 (C), LDB2 vs. SSBP2/3/4 (D), and LDB2 vs. RLIM (E). LDB2 (WT, T83N or P170L) and candidate interactors were tagged with Flag and HA, respectively. HEK293T cells were transfected with the Flag-construct and HA-construct indicated. After Flag-LDB2 was immunoprecipitated with a Flag antibody, the immunocomplexes were analyzed with Western blotting with Flag and HA antibodies. The input samples (input) were also analyzed. Green and black arrowhead indicate positions for Flag-Ldb2 and HA-proteins, respectively.

F Co-immunoprecipitation of endogenous SSBP3 and LMO3 with LDB2. Immunocomplexes from mouse brain lysate using an antibody against LDB2, LMO3, or SSBP3, or a control antibody were analyzed by Western blotting using an anti-LDB2 (left) or anti-SSBP3 (right) antibody.

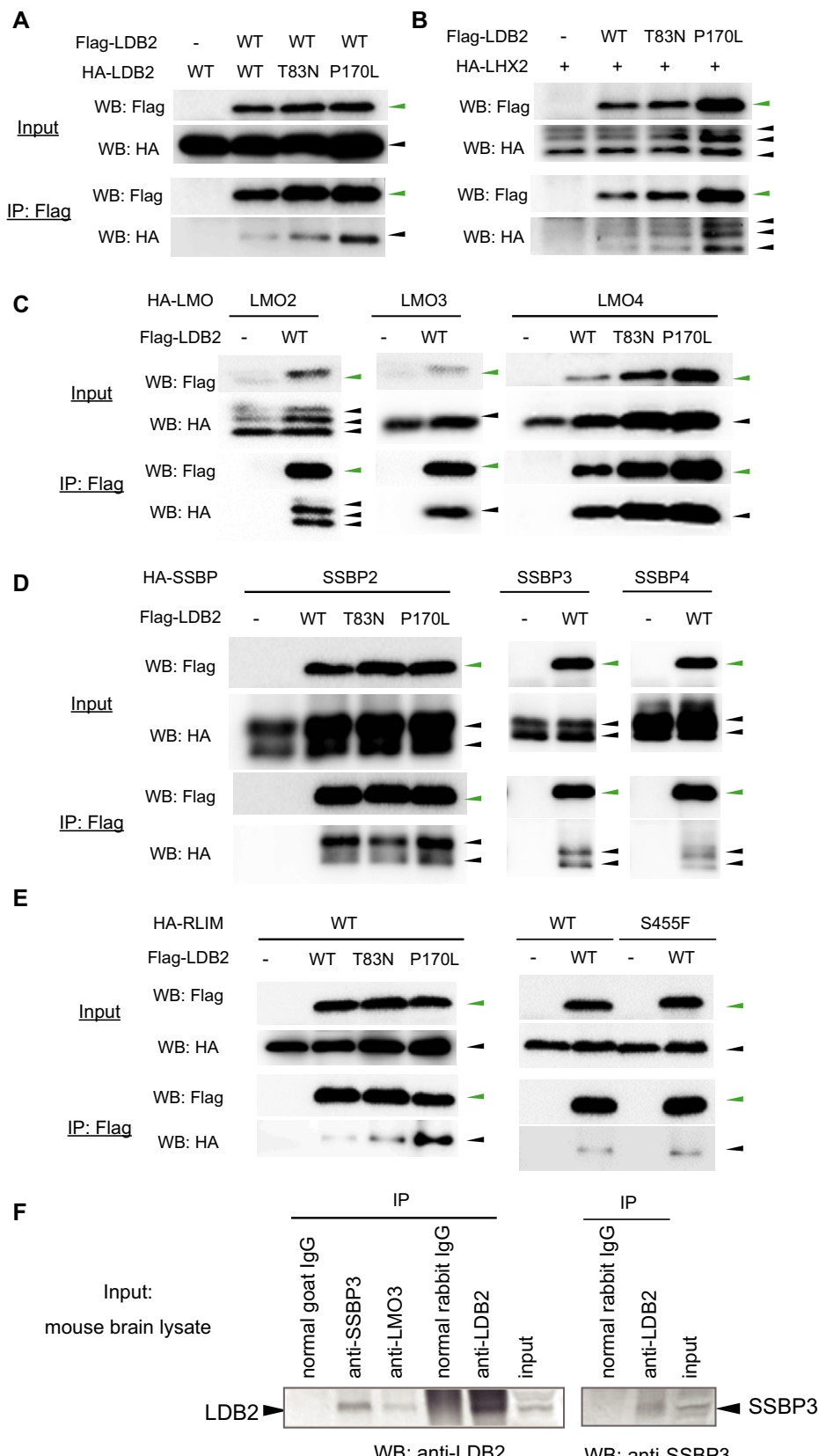

**Figure 7.**

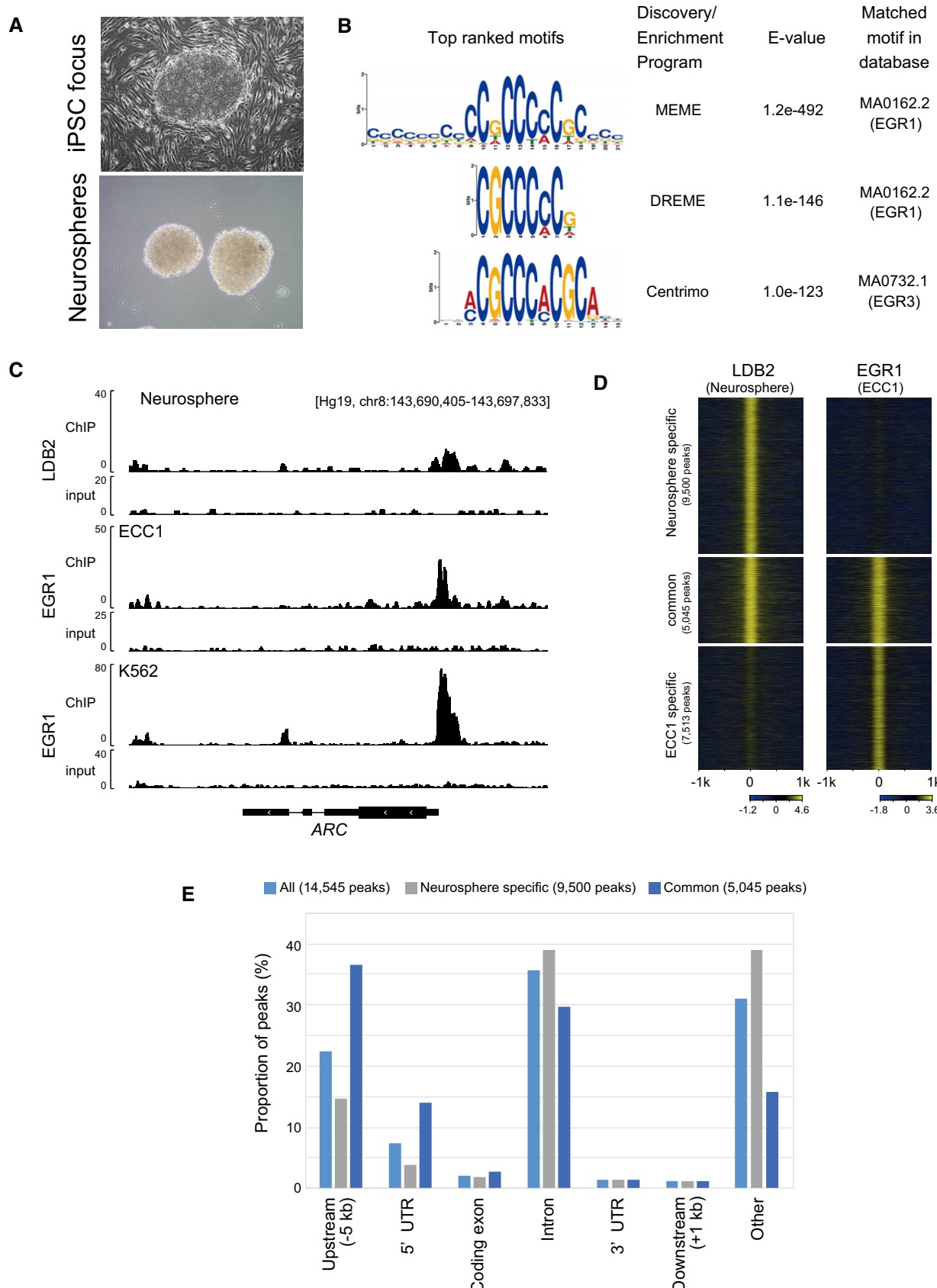

**Figure 8.**

**Figure 8.  Possible cooperation of LDB2 and EGR transcription factors revealed by ChIP-seq.**

A   Human neurosphere (bottom) induced from iPS cells (top) from a healthy control subject.

B   Sequence motifs identified by the MEME-ChIP program. Shown are the top three motifs identified in the LDB2 ChIP-seq peak region (summit $\pm$ 100 bp of the top 2,000 ChIP-seq peaks) (See Fig EV4 for more motifs).

C   Overlap of ChIP-seq peaks of LDB2 in neurosphere cells and ChIP-seq peaks of EGR1 in human cultured cell lines ECC1 and K562, at *ARC* (See Appendix Table S3 for the dataset used for the analysis).

D   Genome-wide pattern of overlap between the ChIP-seq peaks of LDB2 in neurosphere cells and the ChIP-seq peaks of EGR1 in the ECC1 cell line. Fold enrichment (log$_2$) of ChIP-seq reads against the input reads were calculated for each ChIP-seq peak region (summit $\pm$ 1,000 bp) in each category (neurosphere specific, common, and ECC1 specific) and displayed in a heatmap by ngs.plot (Shen *et al*, 2014). See Fig EV4 for the overlap between the ChIP-seq peaks of LDB2 and EGR1 in other cell lines.

E   Annotation of ChIP-seq peaks. ChIP-seq peaks of LDB2 in the neurosphere cells (all, neurosphere specific, and common, shown in D were analyzed by the PAVIS program (Huang *et al*, 2013).

**Table 2.  Gene Ontology genes by ChIP-seq analyses of LDB2 and EGR proteins in Upstream (< 5 kbp) and 5'-UTR region**

| GO biological process complete | *P* value[a] | FDR[b] | Fold Enrichment |
|---|---|---|---|
| Ventricular septum morphogenesis | $1.1 \times 10^{-5}$ | $4.7 \times 10^{-4}$ | 3.4 |
| Regulation of dendritic spine morphogenesis | $1.3 \times 10^{-5}$ | $5.3 \times 10^{-4}$ | 3.2 |
| Segment specification | $9.5 \times 10^{-4}$ | $2.3 \times 10^{-2}$ | 3.9 |
| Cell surface receptor signaling pathway involved in heart development | $1.2 \times 10^{-3}$ | $2.7 \times 10^{-2}$ | 3.3 |
| Regulation of transcription from RNA polymerase II promoter involved in heart development | $1.4 \times 10^{-3}$ | $3.1 \times 10^{-2}$ | 4.0 |
| Establishment of mitochondrion localization | $2.0 \times 10^{-3}$ | $4.2 \times 10^{-2}$ | 3.0 |
| Positive regulation of gene silencing by miRNA | $2.6 \times 10^{-3}$ | $4.5 \times 10^{-2}$ | 3.2 |

The Gene Ontology Resource GO Enrichment analysis (http://geneontology.org) was conducted using a gene set produced by a ChIP-seq overlapping analysis of the LDB2 and EGR proteins. The overlapping peaks detected in the upstream (~5 kbp) or 5'-UTR region were used in the analysis. Biological processes with a fold enrichment of > 3.00 are listed.
[a]Fisher's exact test.
[b]false discovery rate.

The ChIP-seq analyses suggested that LDB2 cooperates with EGR transcription factors in the LDB2-driven gene expression machinery (Fig 9). Prior studies by us and others have suggested a potential role of the *EGR* family genes in the pathogenesis of schizophrenia (Yamada *et al*, 2007; Kim *et al*, 2010; Cheng *et al*, 2012). The LDB-EGR system controls the expressions of many genes, including *ARC/Arc* (Appendix Table S2), whose dysregulation may cause the abnormal spine physiology in *Ldb2* KO mice and schizophrenia (Moyer *et al*, 2015).

It remains to be clarified how LDB2 cooperates with the EGR family proteins to control gene expression, since the EGR family members were not seen in the list of the potential LDB2 interactors by yeast two-hybrid screening. In addition, to our knowledge, the LDB2-binding partners identified in this study are not known as EGR interactors. Although the *Egr3* transcripts show an expression pattern similar to that of *Ldb2* in the mouse cerebral cortex (Allen Brain Atlas: https://mouse.brain-map.org/gene/show/13431), a conventional co-immunoprecipitation assay did not detect an association between LDB2 and EGR3. The interaction between LDB2 and EGRs may be too weak and/or unstable to be detected by conventional methods, or an unidentified protein(s) may be needed for the interaction. The EGR1 protein was rarely detected in the adult mouse brain, making it difficult to test the interaction with LDB2. It is essential to identify the "missing link" between LDB2 and EGRs in the future study to reveal the whole picture of the gene expression control by the LDB2-EGR complex (Fig 9).

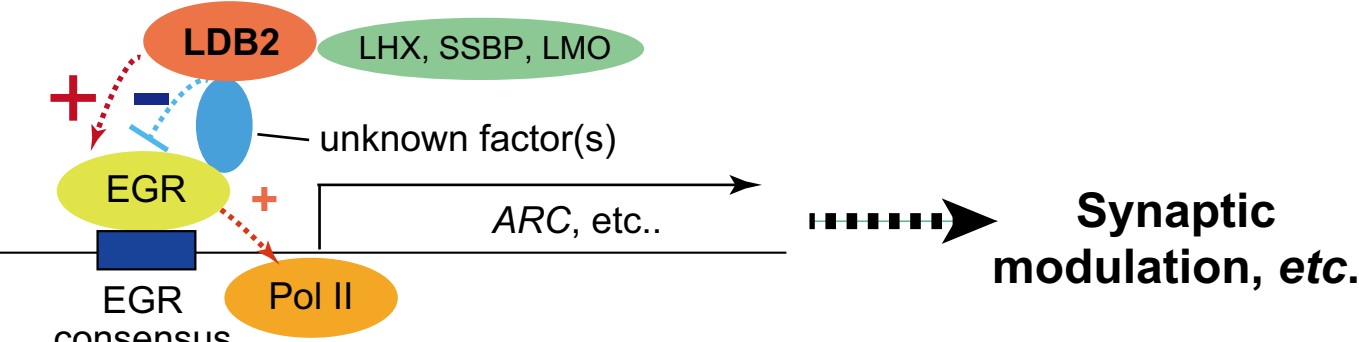

**Figure 9.  A proposed model in which the "LDB2-EGR" axis controls gene expression to regulate synaptic function.**

ChIP-seq analyses identified the EGR family members as a functional partner of LDB2. Our study suggested that LDB2 suppresses (blue) and upregulates (red) EGR-mediated gene expression in a context-dependent manner. See the main text for more detailed information.

In summary, we present multiple lines of evidence that the "LDB2-EGR" axis orchestrates the regulation of gene expression networks involving *ARC* and that a perturbation of this system is potentially related to the pathogenesis of mental disorders, such as schizophrenia and bipolar mania.

# Materials and Methods

### Study approval and ethics issues

All the recombinant-DNA experiments, human-induced pluripotent stem cell (iPSC), and animal studies were performed in compliance with relevant laws and approved by the ethics committees of RIKEN, Tokyo Metropolitan Institute of Medical Science, and the University of Tokyo. Informed consent was obtained from all subjects and that the experiments conformed to the principles set out in the WMA Declaration of Helsinki and the Department of Health and Human Services Belmont Report.

### Mice

The animals were housed in groups of 4–5 in standard cages, in a temperature and humidity-controlled room with a 12-h light/dark cycle (lights on at 08:00). The animals had free access to standard lab chow and tap water. The inbred C57BL/6N (B6N) mice were obtained from Japan SLC (Hamamatsu, Japan). Sperm of the *Ldb2* knockout (KO) mice, where the function of the gene was knocked out by the replacement of exon 1 with the lacZ/neo$^r$-expression cassette, were obtained from the MMRRC (Mutant Mouse Resource & Resource Centers) (RRID: MMRRC_011733-UCD, B6;129S5-*Ldb2*$^{tm1Lex}$/Mmucd). The sperm were subjected to *in vitro* fertilization with oocytes from B6N females. The fertilized eggs were transplanted into the uterus of pseudo-pregnant dams to produce first generation animals. Genotyping was performed using genomic DNA from the tail by multiplex genomic PCR with the three primers: Neo3a (5'-GCAGCGCATCGCCTTCTATC-3'), 0813-4 (5'-AAGAACA CAGCCCATGTGC-3'), and 0813-6 (5'-TCTTGTTCATCTCATAGA TTCG-3'). To obtain the gene KO and wild-type (WT) control mice, heterozygous mice produced by backcrossing the founder animal to the B6N strain for at least six generations were intercrossed. Basically, all animal experiments were done using male at 9–17 weeks after birth.

### Antibodies

A polyclonal antibody against LDB2 was raised to the N-terminal peptide (MSSTPHDPFYSSPFGPFYRRHTPYMVQPEYC) coupled with KLH (keyhole limpet hemocyanin) in rabbits. The thiol residue of the C-terminal cysteine was used for crosslinking the peptide to KLH. The anti-serum was used for the Western blot analysis (dilution, 1:250). For ChIP-seq experiments, an anti-LDB2 antibody was affinity-purified from the anti-serum with the beads coupled with the immunogen peptide. A monoclonal antibody was raised by immunizing rats with the same KLH-peptide according to the previously described method (Kishiro *et al*, 1995). The clone named 4-1F was used for immunohistochemistry (ascites, dilution 1:50). In our tests, some of commercially available antibodies produced extra discrete bands around an expected molecular mass of LDB2 in Western blot analyses of the tissue lysates from the *Ldb2* KO animals, and therefore, they were non-specific.

### Behavioral analysis

#### Open-field test
The mice behavior was recorded in the open-field (500 × 500 × 300 mm; W × D × H; gray acrylic walls, bright-light conditions at 70 lux) for 10 min. The automatic monitoring system Time OFCR4 (O'Hara & Co., Ltd.) was used to assess the total distance travelled for 10 min and the percentage of time spent in the center area (size: 25 % of the field).

#### Homecage activity test
The locomotor activity was measured in the homecage (196 × 306 × 166 cm; W × D × H) with an infrared sensor on the ceiling (Supermex; Muromachi Kikai). The data were accumulated at 10-minute intervals. The mice were monitored for 3 days in a 12-h light/dark cycle.

#### Tail suspension test
An automated tail suspension apparatus with four channels was used to measure immobility during a 10-min session. The mice were suspended from a hook by the tail using non-irritating adhesive scotch tape. Immobility was analyzed using a charge-coupled device (CCD) camera and Image J TS4 (O'Hara & Co., Ltd.).

#### Y-maze test
Y-maze was prepared from gray acryl and comprised three arms (arm length, 40 cm; arm bottom width, 3 cm; arm upper width, 10 cm; the height of the wall, 12 cm). The experiment was started by placing the mice in the center of the Y-maze field. A modified version of the Time YM2 program (O'Hara & Co., Ltd.) was used to record the number of entries and alterations. The data were collected for 7 min.

MK-801/Methylphenidate-induced locomotor hyperactivity test: The mouse tested was moved from home cages to a test cage, 1 day before the examination, and then habituated to the new environment. Recordings of locomotor were started 1 h prior to the MK-801 or methylphenidate injection. The mice were then briefly removed from the cage and injected subcutaneously with MK-801 (0.2 mg/kg body weight) or methylphenidate (MPD; 3 mg/kg body weight). The mice were immediately returned to the same test cage, and the locomotor responses to MK-801 or MPD were monitored for 2.5 h. Locomotor activity was measured using an infrared sensor (Supermex; Muromachi Kikai, Tokyo, Japan).

Other behavioral analyses were conducted by following the protocols already published from our group [Wada *et al*, 2020, Maekawa *et al*, 2017, Shimamoto *et al* 2014, Ohnishi *et al*, 2013].

### Neurogenesis assay in the hippocampus

The measurement of neurogenesis was conducted as described elsewhere. In brief, mice (three each for WT and KO mice) at 5 weeks of age were intraperitoneally injected with 100 mg/kg BrdU (bromodeoxyuridine) three times with intervals of 2 h to label cells in the S phase of cell cycle. Twenty-four hours after the final injection,

the animals, after being deeply anesthetized with pentobarbital, were fixed with 4% paraformaldehyde in 1x phosphate buffer *via* the heart. Serial cryosections (14-μm coronal section) were prepared from the postfixed brains, and the sections were stained with an anti-BrdU antibody labeled with a fluorescent dye. The BrdU-positive nuclei were visualized using a fluorescence microscope (Olympus, Tokyo, Japan), and BrdU$^+$ cells were counted.

**Yeast two-hybrid screening**

The yeast two-hybrid screening was conducted using ProQuest Two-Hybrid System (Invitrogen, Grand Island, NY, USA) to identify LDB2-binding proteins, following the manufacture's instruction. In brief, the full-length human LDB2 cDNA was inserted into pDEST32 and the resultant plasmid was used as a bait construct. The adult human brain cDNA library (Invitrogen), which was constructed in pEXP-AD502, was used. Both the bait and library were transformed into the MaV203 yeast strain. Primary screening was done on Leu$^-$/Trp$^-$/His$^-$-plates containing 3-amino-1,2,4-triazole (3AT). Resultant His$^+$ transformants were further selected using the X-gal assay and grown on Leu$^-$/Trp$^-$/Ura$^-$-plates to identify cells expressing *lacZ* and *URA3*. Competent *E. coli* cells were transformed with plasmid DNA obtained from candidate clones. *E. coli* clones containing both the bait and prey constructs were selected with antibiotic (ampicillin and gentamicin) resistance. The isolated prey plasmids were subjected to base sequencing and the BLASTn (https://blast.ncbi.nlm.nih.gov/Blast.cgi) analysis in the NCBI database.

**Co-immunoprecipitation assay**

Plasmid construction, site-directed mutagenesis, and a co-immunoprecipitation assay were performed as described elsewhere (Yamashita *et al*, 2001; Ohnishi *et al*, 2003) with minimum modifications. In brief, each cDNA indicated was inserted into pCMV-HA (Takara Bio Inc., Shiga, Japan) or pCMV-FLAG (Takara Bio Inc.). Site-directed mutagenesis was performed to introduce the mutations using a conventional method. HEK293T cells were transfected with an indicated combination of the FLAG and HA constructs to co-express FLAG-tagged and HA-tagged proteins. After incubation for 48 h, the cell lysates were prepared in the lysis buffer (25 mM Tris–HCl, pH 7.8, 125 mM NaCl, 1% Triton X-100, 0.5 mM EDTA, 1 mM PMSF, 2 μg/ml aprotinin), and they were precleared by centrifugation at 15,000 x *g* for 15 min. Flag-LDB2-containing complexes in the precleared lysate were captured with anti-Flag antibody (Sigma-Aldrich, St. Louis, MO, USA)-preabsorbed Protein G-Sepharose beads (GE Healthcare, Chicago, IL, USA) by gentle rotation for 3 h. The immunocomplexes were washed with the lysis buffer four times and analyzed with Western blotting using anti-Flag and anti-HA antibodies (Roche Diagnostics, Rotkreuz, Switzerland).

**Analysis of neuronal activity in the amygdala**

To analyze behaviorally induced neural activity in the *Ldb2* knockout mice, we crossed the *Ldb2* knockout mice with homozygous transgenic mice expressing Venus protein under control of the full-length *Arc* promoter (Arc-Venus) (Mikuni *et al*, 2013; Vousden *et al*, 2015), which was destabilized by inserting the PEST sequence, to yield Arc-Venus tg/tg, *Ldb2*$^{-/-}$ double mutant mice (AV-KO) and

control littermates (AV-WT). To analyze Venus fluorescence, AV-KO and AV-WT (10–14 weeks old) were moved to experimental room at least three hours before the fear-conditioning trial (Fig EV4C). Mice were divided into three groups, depending on their conditioning trials. In the "Paired" group, mice were placed in the conditioning chamber for 2 min and eight tone (70 dB, 10 s)-shock (0.35 mA, 2 s) pairings were delivered during the following 7 min. One minute after the last tone-shock pairing, mice were returned to their home cage. In the "Unpaired" group, mice were delivered eight tones and shocks separately during the same period. The "Naïve" group received no treatment (kept in a home cage) in the experimental room. Three hours after the behavioral trials (Fig EV4C), the mice were deeply anesthetized by intraperitoneal (i.p.) injection of tribromoethanol and transcardially perfused with 20 mL of phosphate-buffered saline (PBS, pH 7.4), followed by 50 ml of 4% formaldehyde in PBS. The skulls were dissected, and the brains were further fixed in 4% formaldehyde in PBS overnight at 4°C. The brains were then transferred first to 15% sucrose in PBS for 8 h, then to 30% sucrose overnight for cryoprotection, and were then frozen in the O. C. T. compound (Tissue-Tek/Sakura, Tokyo, Japan). To analyze Venus fluorescence, three coronal sections including the amygdaloid complex (between Bregma −1.855 mm and −2.155 mm) were collected with a freezing microtome (90 μm in thickness; Polar D, Tissue-Tek, Japan). To analyze immunofluorescence staining with Venus fluorescence, 20-μm coronal cryosections were collected serially and were stored in PBS. Free-floating sections were incubated in 0.3% Triton X-100 for 30 min. The sections were pre-incubated with blocking PBS buffer containing 0.3% Triton X-100 and 10% normal goat serum (GIBCO, Co Dublin, Ireland) for 1 h at room temperature. Then, the sections were incubated first with an anti-CaMKIIα rabbit monoclonal primary antibody (Abcam #ab92332) in blocking buffer for 48 h at 4°C and were then washed with PBS containing 0.3% Triton X-100. Next, they were incubated with Alexa Fluor 594-conjugated anti-mouse secondary antibodies (1:5,000, Invitrogen) in blocking buffer for 2 h at room temperature. The staining protocol for anti-GAD67 (Millipore #MAB5406; Millipore, Burlington, MA, USA) immunofluorescence was as follows: The sections were pre-incubated with blocking PBS buffer containing 10% normal goat (GIBCO) serum for 1 h at room temperature. Then, they were incubated with primary antibodies in blocking buffer for 48 h at 4°C. The sections were then washed with PBS and were incubated with Alexa Fluor 594-conjugated anti-mouse secondary antibodies (1:5000, Invitrogen) in blocking buffer for 2 h at room temperature. The sections were then washed with PBS and mounted on slides, cover-slipped with DAPI fluoromount-G (Southern Biotech, Birmingham, AL, USA), and sealed. For fluorescence measurements, we used a fluorescence microscope with an optical sectioning function (ApoTome2; Zeiss, Jena, Germany) equipped with Plan-Apochromat 10×/0.45 to analyze Venus fluorescence singly, and Plan-Apochromat 20×/0.8 to analyze immunofluorescence with Venus. The acquired images were exported as TIFF images. We detected Venus-positive neurons with their fluorescence intensities not less than 15 arbitrary units (AU) using the particle analysis of NIH Images software (http://imagej.nih.gov/ij/). We analyzed open-source cell segmentation software CellProfiler (Carpenter *et al*, 2006) to analyze Venus-positive cells with immunofluorescence.

For visualized patch-clamp experiments, AV-KO and AV-WT mice underwent fear conditioning in the same way as the Paired group of the Venus fluorescence analysis. One hour after the conditioning, the mice were anesthetized deeply with halothane and decapitated, and then, the brains were removed. Coronal brain slices (400-μm thick) were cut with a tissue slicer (Leica 1200S; Leica, Wetzlar, Germany) in the cutting solution containing 2.5 mM KCl, 1.3 mM $NaH_2PO_4$, 25 mM glucose, 20 mM HEPES, 30 mM $NaHCO_3$, 5 mM (+)-sodium L-ascorbate, 92 mM N-methyl-D-glucamine, 12 mM N-acetyl-L-cysteine, 2 mM thiourea, 0.5 mM $CaCl_2$ and 10 mM $MgSO_4$, and saturated with 95% $O_2$ and 5% $CO_2$. The slices were incubated at 35.6°C for 15 min in the slice cutting solution, then incubated for at least 1 h at room temperature in the submersion-type holding chamber filled with the Krebs-Ringer solution containing 119.0 mM NaCl, 2.5 mM KCl, 1.0 mM $NaH_2PO_4$, 2.5 mM $CaCl_2$, 1.3 mM $MgSO_4$, 26.2 mM $NaHCO_3$ and 11.0 mM glucose, and saturated with 95% $O_2$ and 5% $CO_2$, and then, a slice was transferred to the recording chamber, where the solution was perfused at 1.5–2.0 ml/min at 25°C. Whole-cell patch-clamp recordings were made from Venus-positive or -negative cells in the lateral amygdala (LA). A recording patch pipette (3–7 MΩ) was filled with the internal solution containing 122.5 mM Cs-gluconate, 17.5 mM CsCl, 10.0 mM HEPES, 0.2 mM EGTA, 2.0 mM MgATP, 0.3 mM $Na_3GTP$, and 8.0 mM NaCl (pH 7.2, 295-300 mOsm). A bipolar stainless stimulating electrode was placed in the ventral striatum just medial to the LA to stimulate fibers originating in the auditory thalamus. Cells were held at −80 mV to record EPSCs and mEPSCs with an Axopatch 1D (Molecular Devices, Foster City, CA, USA) amplifier, and the signal was digitized with Digidata 1440A (Molecular Devices, San Jose, CA, USA). Picrotoxin (100 μM) was present in the Krebs-Ringer solution, and tetrodotoxin (1 μM) was added to record mEPSCs. Differential interference-contrast and Venus fluorescence images of all the recorded cells were photographed in the auto mode using an IR-1000 camera (DAGE MTI, Michigan City, IN, USA) and were captured and saved with video capture GV-USB2 (I-O DATA, Kanazawa, Japan). mEPSCs were analyzed using the Mini Analysis (6.0.7) software (Synaptosoft, Decatur, GA, USA). Of all the events detected, only the events that did not overlap with other events were accepted for mEPSC amplitude analysis, whereas all the events were included in the frequency analysis. Significant differences between the groups were evaluated with two-way ANOVA. Fluorescence ratios of cell-body area versus their background area were regarded as their relative fluorescence intensities.

## Chromatin immunoprecipitation-sequencing (ChIP-seq) analysis

Neurospheres differentiated from iPSCs that have been established from a healthy control (ID: HPS0063, obtained from RIKEN BioResource Research Center) (Takahashi et al, 2007) were used for the ChIP experiment. ChIP-seq data of the EGR1 were obtained using cancer-derived cell lines, ECC1 (endometrial cancer), HCT116 (colon cancer), and K562 (erythromyeloblastoid leukemia), and using an ES cell line H1ES from the NCBI SRR. After differentiating the iPS cells to neurosphere following the standard method, cells were fixed in a PBS (-) solution containing 1 % formaldehyde and 2 mM DSG (disuccinimidyl glutarate) for 10 min at room

### The paper explained

#### Problem

The pathogenesis of mental disorders including schizophrenia remains largely unknown. Our group reported a schizophrenia patient who harbored a balanced chromosomal translocation $t(4;13)(p16.1;q21.31)$ and recently identified the LDB2 (Lim Domain-binding 2) gene in the breakpoint at chromosome 4 by next-generation sequencing. However, a potential role of the LDB2 gene, which encodes a putative transcription regulator without DNA-binding domains, in disease mechanism remains elusive.

#### Results

Ldb2 knockout (KO) mice displayed multiple behavioral abnormalities including hyperactivity, enhanced sensitivities to a psychostimulant and a hallucinogenic drug, and deficits in fear-conditioning test. Some of these were alleviated by the treatment with an antipsychotic or with a mood stabilizer lithium, indicating that the Ldb2 KO mice satisfy both face validity and predictive validity as a model for schizophrenia and bipolar mania. Molecular analyses revealed that LDB2 binds to multiple transcription factors and transcription regulatory factors and that Ldb2 regulates the expression of immediate early gene Arc in the amygdala. ChIP-seq analyses suggested that LDB2 cooperates with the EGR transcription factors, whose genes are associated with schizophrenia, to control synaptogenesis-related genes including Arc/ARC. These results suggest that the LDB2-EGR signaling cascade has a role in psychiatric diseases including schizophrenia and bipolar disorder.

#### Impact

This study revealed a potential role of transcription factor cooperation, the "LDB2-EGR axis", in the pathogenesis of schizophrenia and bipolar disorder, providing a novel insight into the mechanism of mental disorders.

temperature. The crosslinking reaction was quenched by adding glycine to the final concentration of 125 mM. Chromatin immunoprecipitation (ChIP) was performed as previously described (Kadota et al, 2017) except that, RIPA buffer was used instead of LB3 lysis buffer at permeabilization and sonication steps. Chromatin lysate was prepared from $1 \times 10^7$ cells in 1 ml of RIPA buffer, using the Covaris E220 sonicator (Covaris, Woburn, MA, USA) and a milliTUBE under the following condition; PIP 175, 10% duty factor, 200 cycles per burst, 7°C bath temperature, and 55 min of duration. Immunoprecipitation was performed using 100 μl of Protein A beads bound with 20 μg of the anti-LDB2 polyclonal antibody with 1 ml of the chromatin lysate in the RIPA buffer, supplemented with 1 mg/ml of BSA and 1 x protease inhibitor cocktail. Illumina compatible libraries were prepared from 20 ng of input DNA or the ChIP DNA obtained from $1 \times 10^7$ cells, using the KAPA LTP Library Preparation kit (KAPA Biosystems, Wilmington, MA, USA). The libraries were sequenced on Illumina HiSeq1500, using the HiSeq SR Rapid Cluster Kit v2-HS and the HiSeq Rapid SBS Kit v2-HS (Illumina, San Diego, CA, USA), to obtain single-end 80 base sequencing reads. Trimming of the reads at the adapter sequence or low-quality bases was performed by TrimGalore v0.4.1. Mapping of the reads to the human hg19 genome was performed by Bowtie v1.1.2 using "-m 1 -n 2 -a --best --strata" options to obtain only uniquely mapped reads. Peak calling was performed by MACS2 v2.1.1 with the default parameters. Sequence motifs enriched inside the LDB2 ChIP-seq

peaks were identified by MEME-ChIP v4.11.1 with the JASPAR database (JASPAR_CORE_2016_vertebrates.meme; (Fornes *et al*, 2020)), using the nucleotide sequences of the peak summit $\pm$ 100 bp of the top 2000 ChIP-seq peaks ordered by the fold enrichment values of MACS2. ChIP-seq data for the EGR1 protein of the cell lines, ECC1 (SRR577960, SRR577961, SRR577989, SRR577990), H1ES (SRR577840, SRR577841, SRR351691, SRR351692), HCT116 (SRR577956, SRR577957, SRR577901, SRR577902), and K562 (SRR578001, SRR578002, SRR578003, SRR578004, SRR351661, SRR351662), were obtained from NCBI SRA and analyzed in the same way as for the LDB1 protein of neurosphere. Overlap of the ChIP-seq peaks of the different datasets was identified by the intersect feature of Bedtools v2.19.1, and the Venn plot was generated by the VennDiagram package in R. Heat map of the ChIP-seq reads at $\pm$ 1,000 bp region of the ChIP-seq peak summits was generated by ngs.plot (Shen *et al*, 2014). Annotation of ChIP-seq peaks was performed by PAVIS (Huang *et al*, 2013) using the UCSC Known Genes of human hg19 as the gene set with the search distance parameter set to 0.

## Data availability

LDB2 ChIP-seq data have been deposited at DDBJ under the accession number DRA010493 (https://ddbj.nig.ac.jp/DRASearch/submission?acc = DRA010493).

**Expanded View** for this article is available online.

## Acknowledgements

We thank Dr. T. Kato (RIKEN CBS) for kindly offering us unpublished information on the *RLIM* variant in the bipolar patient, Drs. T. Kimura (National Center for Geriatrics and Gerontology, Japan) and H. Furudate (Saitama University) for useful suggestions, and Ms. C. Kimura for technical support. We are also grateful to Lexicon Pharmaceuticals and the MMRRC for providing us with a sperm stock of *Ldb2* KO mouse. We are deeply thankful to the members of the Research Resources Division of RIKEN CBS for their technical support on peptide synthesis, production of polyclonal antibodies in rabbits, DNA-sequencing service, animal care, and mouse embryo manipulation, and to the members of the Laboratory for Phyloinformatics, RIKEN BDR for the NGS sequencing. This research was supported by JSPS KAKENHI under Grant Numbers 16K07017, 24591737, 21591535, and 20K07934 (to T. O.), 15K01848 (to Y. K.), 17790834 (to M. A.), and 23220008, 25116505, 19H03321 and 19H04876 (to T. M.), and by the Grant-in-Aid for Scientific Research on Innovative Areas from the MEXT under Grant Numbers JP18H05435 (to T. Y.). The funding agencies had no influence on study design, data collection, data analysis, interpretation, or writing of the report.

## Author contributions

TO, TM, and TY designed the studies. TO, YK, FA-Y, MK, TI, KY, AW, HO, KTa, YH, YI, MT, IO, CS-M, MMa, MA, MMi, KTo, YN, RK, KS, and AY conducted the experiments and/or acquired the data. TT, HO, and HB provided valuable materials. TO, YK, FA-Y, MK, AN, SB, TH, MI, SK, TM, and TY contributed to the analysis and/or interpretation of the results. TO, YK, FA-Y, MK, TM, and TY wrote the manuscript.

## Conflict of interest

The authors declare that they have no conflict of interest.

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
