## [Review Process File · EMBO Molecular Medicine]

Cooperation of LIM domain-binding 2 (LDB2) with EGR in the pathogenesis of schizophrenia

Tetsuo Ohnishi, Yuji Kiyama, Fumiko Arima-Yoshida, Mitsutaka Kadota, Tomoe Ichikawa, Kazuyuki Yamada, Akiko Watanabe, Hisako Ohba, Kaori Tanaka, Akihiko Nakaya, Yasue Horiuchi, Yoshimi Iwayama, Manabu Toyoshima, Itone Ogawa, Chie Shimamoto-Mitsuyama, Motoko Maekawa, Shabeesh Balan, Makoto Arai, Mitsuhiro Miyashita, Kazuya Toriumi, Yayoi Nozaki, Rumi Kurokawa, Kazuhiro Suzuki, Akane Yoshikawa, Tomoko Toyota, Toshihiko Hosoya, Hiroaki Okuno, Haruhiko Bito, Masanari Itokawa, Shigehiro Kuraku, Toshiya Manabe, and Takeo Yoshikawa

DOI: [10.15252/emmm.202012574](https://doi.org/10.15252/emmm.202012574)

Corresponding author(s): Tetsuo Ohnishi (tetsuo.ohnishi@riken.jp) , Toshiya Manabe (tmanabeky@umin.ac.jp), Takeo Yoshikawa (takeo.yoshikawa@riken.jp),

Review Timeline:

Submission Date:	6th Jul 20
Editorial Decision:	6th Aug 20
Revision Received:	30th Nov 20
Editorial Decision:	4th Jan 21
Revision Received:	21st Jan 21
Accepted:	25th Jan 21

Editor: Jingyi Hou

Transaction Report:

Thank you for the submission of your manuscript to EMBO Molecular Medicine. We have now received feedback from the three referees whom we asked to evaluate your manuscript. As you will see from the reports below, the referees acknowledge the potential interest of the study. However, they also raise substantial concerns about your work, which should be convincingly addressed in a major revision of the present manuscript.

While referee #1 and referee #2 are more supportive of publication, referee #3 is much more reserved and mentions limitations that must be addressed for the study to be conclusive: additional controls and experiments are required, the inconsistencies between data and text need to be addressed, along with clarifications and details. Referee #3 is also concerned about several potential overstatements, which we would ask you to pay attention to and address within reason.

We would welcome the submission of a revised version within three months for further consideration.

***** Reviewer's comments *****

Referee #1 (Remarks for Author):

Review of

Cooperation of LIM domain binding 2 with EGR transcription factors to regulate gene expression networks relevant to mental disorders

Ohnishi et al studied a role of LIM domain binding 2 (LDB2) gene encoding a putative transcription regulator in mental disorders. In the present study, they used different approaches, phenotypic analyses of Ldb2 mouse model; yeast two-hybrid, identification of direct genomic targets of Ldb2 by chromatin immunoprecipitation (IP) followed by sequencing (ChIP-seq).

The present manuscript deserves publication with minor revisions.

Minor revision in particular to mouse model data

Mouse model data: pp 9-15 and Figures 2-6.

For non-specialist reader, I think that it can be useful to resume phenotypical traits that characterize bona fide mouse models of schizophrenia. These phenotypical traits include neuronal architecture, synaptic plasticity, circuitry deficits and behavioral paradigms dependent on learning such as spatial working memory (WM) performance using a T-maze delayed non-match to place (DNMTP) task; Pre-Pulse Inhibition (PPI) (see for instance Gogos et al., 2019; Mukai et al., 2019). This information can highlight translational phenotypes from mouse to humans.

Pages 11-14: Analysis of Arc expression as a function of a cognitive test.

The authors generated Arc Venus tg/tg; Ldb2 KO (AV KO), and Arc Venus tg/tg; Ldb2 WT (AV WT) to monitor neuronal activities in the amygdaloid complex through the Venus fluorescence.

They found that the increment of the Middle Venus positive neurons was significantly larger in the AV KO compared to AV WT. These results indicating that Ldb2 suppresses the Arc gene transcription responding to the tone shock paired stimuli in the LA neurons (Figure 5). Synaptic activity is reduced as a function of Arc expression whatever the genotype (Figure 6 and text page 14).

The logic is to present the result (Figure 6) before the analysis of the number of positive neurons (Figure 5), in order to present first the relation between arc expression and synaptic activity and second the inverse relationship between the number of ARC expression and the level of Ldb2 protein.

Furthermore, I think that indication of regression in graphs of Figure 6 with a graph for each condition will increase the readability of the figure.

Discussion

Phenotypic traits of the mouse model need to be discussed in comparison to what found in bona fide mouse models

Results of Figure 6 have to be discussed with the indication that the decrease in mEPSPs possibly reflect a change in the number of postsynaptic AMPA receptors.

Small corrections:

Figure 1A Dimerization instead of Dimerization

References

Gogos, J.A., Crabtree, G., and Diamantopoulou, A. (2019). The abiding relevance of mouse models of rare mutations to psychiatric neuroscience and therapeutics. *Schizophr. Res.* Published online April 12, 2019. <https://doi.org/10.1016/j.schres.2019.03.018>.

Mukai J, Cannavò E, Crabtree GW, Sun Z, Diamantopoulou A, Thakur P, Chang CY, Cai Y, Lomvardas S, Takata A, Xu B, Gogos JA. Recapitulation and Reversal of Schizophrenia-Related Phenotypes in *Setd1a*-Deficient Mice. *Neuron*. 2019 Nov 6;104(3):471-487.

Referee #2 (Comments on Novelty/Model System for Author):

The authors present an extensive modeling study using both mouse and iPSC-derived neurospheres to examine the potential for a novel risk gene for schizophrenia. An alternative cellular model would be iPSC-derived neurons as well as iPSC-derived neural cells from the patient.

Referee #2 (Remarks for Author):

Ohnishi and colleagues conducted a very extensive mouse study to investigate the potential role of LDB2 in brain function and behavior. In a prior study, this group identified a rare translocation t(4;13) (p16.1;q21.31), in a schizophrenia patient. While the breakpoint on chromosome 13 was devoid of nearby genes, the breakpoint at chromosome 4 localized at the upstream region of the LDB2 gene, in effect, rendering the latter a plausible candidate deserving further studies. Toward this end, the authors created a knockout (KO) *Ldb2* mouse, examined gene expression brain, conducted multiple behavioral assays and showed phenotype rescue using neuropsychiatric drugs. Several protein interactors with LDB2 were detected following yeast two-hybrid assay in human embryonic brain cDNA library. ChIP-seq analysis in human iPSC-derived neurospheres showed enrichment of EGR target DNA elements. The authors state that these findings suggest the possible involvement of the LDB2-EGR axis in neuropsychiatric disorders.

Comments

This manuscript reports on a tour-de-force effort to use both animal and human cellular models to examine various functional aspects of LDB2, a gene located close to the breakpoint on 4p16, a part of a very rare chromosomal structural aberration in a schizophrenia patient. The findings presented are important and additional supportive future data would heighten the relevance of the LDB2-EGR system to risk for schizophrenia.

Minor comments

- If the parents of the proband were available, were they unaffected and did any of them carry the translocation?
- Are there any known GWAS hits on 4p16.1 & 13q21.31 for any neuropsychiatric disorder?
- Did the schizophrenia genomic DNA carry any of the known pathogenic CNVs, e.g., on 15q, 16p or 22q?
- How far is LDB2 from the 4p breakpoint?
- Did the authors try to sequence the 2 bands on Western blot?
- Why did the authors use iPSC-derived neurospheres instead of either iPSC-derived neural

progenitor cells or neurons?

- It would have been interesting also to use iPSC-derived neural cells from the patient.
- Y-2-H tends to have problems with signal-to-noise ratio, any comments about this issue?
- There are too many numbers in pages 12-14 that are distracting. Presenting these numbers in Tables would help.
- Fig. 1A: Replace "dimerization" with "dimerization"
- Fig. 1D, F & G: An improvement in color intensity would help.
- Fig. 2: Indicate in the legend what blue and red lines represent.
- Fig. 4A: Replace "venral" with "ventral"
- Figs. 5 & 6: Revise the title to indicate what the figures are showing in a manner similar to how the title in Fig. 4. is constructed.
- Fig. 7: Include a legend.
- Fig. 8: The legend can be made more descriptive. Fig. 8F legend is missing.
- Fig. 10: The model can be simplified. You may need to present just the top figure, revised to reflect both activation & repression

Referee #3 (Comments on Novelty/Model System for Author):

Use of KO mice to study the function of the gene (LDB2) is appropriate. However, to insist that the model is a mental disorder model is not appropriate. They need to modify their statement in the manuscript.

Referee #3 (Remarks for Author):

Ohnishi et al. have investigated the expression pattern of Ldb2 in mice, effect of Ldb2 KO in mouse behaviours and Arc expression and identified direct interactors of LDB2 and potential indirect binding sites in DNA. They have postulated possible mechanisms of phenotypes seen in the KO mice and the action of LDB2.

This is the first study generating and studying Ldb2 KO mouse and in that sense, this study is novel and provides data that could benefit others.

However, the link between the schizophrenia and the gene is very weak as it is based on one case although de novo, hence linking the findings of this study to mental disorders, as was stated in several places throughout the manuscript by the authors, is not appropriate and over-stretching. Another point that has to be corrected is the interpretation of their data and the building of hypothesis and model. The manuscript shows either no supporting evidence or rather opposite evidence to build their hypothesis, especially around Arc expression and mEPSCs. Also it has to be noted that mEPSC recording is not a measure of synaptic plasticity in this experimental setting. Often the statements were not directly supported by the results, therefore, either new experimental data needs to be added to support their statements or they need to be modified to reflect the data.

Detailed comments are below.

Introduction

1. Do patient derived tissues (any somatic tissue? Biopsy?) show reduced expression of LDB2? In order to support the breakpoint is affecting the expression of LDB2, this data is needed.

First result section: Ldb2 is expressed in neurons of restricted regions in the brain

2. Has the lab-made LDB2 antibody tested for cross reactivity for LDB1?

3. Fig 1D doesn't necessarily show that Ldb2 is expressed only in neuron. Remove referring of Fig.1D in the first result section claiming this.
4. Fig 1D doesn't show that layer V has most prominent expression. Layer II/III also has strong expression too. Reflect this in the result. Proper quantification is needed with together with NeuN staining in all layers.
5. Fig 1E doesn't prove that Ldb2 expression is highest in layer V. Do not refer Fig 1E when claiming this.
6. When claiming in the expression in Amygdala, there is no evidence showing the expression is only in neurons. Should do co-staining with NeuN to claim this.

Second result section: Ldb2 KO mice display pleiotropic behavioral abnormalities related to mental disorders

7. In the first line "dysregulated expression of LDB2" should be lack of LDB2 as the model they use is deficiency rather than dysregulation.
8. "Although Ldb2 showed a layer-specific expression in the cerebral cortex," This was not investigated fully. The authors need to quantify the % of neurons expressing LDB2 in each cortical layer to support this. Even in the representative picture in Fig 1D, it doesn't seem it has specific expression in layer V.
9. In Fig2, write the number of mice used in each test either in the figure itself or in the figure legend. Same goes for the rest of figures.
10. Details of the behavioural test protocol needs to be written in the supplementary information or method.
11. Add statistical results in Fig 3A and B bar graphs in comparing WT/saline vs WT/MPD and WT/saline vs WT/MK-801.
12. Fig3A bar graph Y axis label is wrong.
13. Explain why the dose of MK-801 went up from 0.2 mg/kg in Fig 2B locomotion test into 0.5 mg/kg in 'popping' behaviour test. Did 0.2 mg/kg MK-801 not produce statistically significant result in popping behaviour? If so, this needs to be mentioned.
14. Looking at Fig 2B open field test, KO mice are hyperactive. However, in Fig 3A and B, the locomotion between KO and WT before the injection (effectively same as Fig 2B open field test then) does not show hyperactivity even in the first 10 mins. This raises a question if the increased locomotion in open field test (Fig 2B) was by chance rather than consistent observation.
15. Fig4A Haloperidol graph has unnecessary stars above WT saline. Remove them.
16. Was the locomotion between WT and KO after saline injection significantly different? There is no significance recorded in the figure so it is assumed that were not different. Then again as commented above (14), the increased locomotion in open field test (Fig 2B) was by chance rather than consistent observation.
17. "Importantly, deficits in fear-conditioning test as well as the hyperlocomotive trait of the KO mice were ameliorated by the treatment with typical (haloperidol) and atypical (clozapine) antipsychotics (Fig. 4A, B)." This sentence is not correct. Fig 4B Haloperidol result doesn't show significant improvement in KO after haloperidol injection. Reflect this in the text. In addition, in both open field and fear conditioning test, both the WT and KO responded to drugs. Therefore, authors cannot justify the suitability of LDB2 KO as schizophrenia model based on this. If so, even the WT would've been considered as the disease model as they have responded to the drug.
18. Add individual mouse data into the bar graphs in Fig 4 as was done in other figures (Fig 2, 3).
19. Fig2A shows significantly increased activity of KO mice in home case both in dark and light phases. In figure 4C, KO has higher activity during dark phase in no lithium carbonate group; however, in the light phase there is no activity difference between WT and KO without lithium carbonate. How do you explain this? Why KO show no hyperactivity in Fig4C in light phase whereas there is a significant hyperactivity in Fig2A in light phase?

20. For contextual test, there isn't haloperidol result. Why is that? Was it not performed or was it not significant?

Third result section: Amygdala of Ldb2 KO mice displays aberrant neuronal response to unconditioned/conditioned stimuli

21. How many coronal sections were used for counting venus+ cells? Please specify.

22. "that Ldb2 of excitatory neurons suppressed the number of the Arc gene-expressing neurons in response to the tone-shock paired or unpaired stimuli in the LA." This is not correct. Fig5B LA graph shows a significant increase of Venus+ cells (Arc expressing cells) in WT paired group. This means that Ldb2 in excitatory neurons did not suppress the expression Arc. If that was the case, then the WT paired group would show no increased in venus + cells in comparison to naïve and unpaired WT groups. The conclusion from Fig5B LA graph needs re-writing to accurately reflect the result. Suggestion: Lack of Ldb2 expression in excitatory neurons in LA of amygdala promoted the expression of Arc upon stimuli during fear conditioning trials.

23. "These results indicate that Ldb2 suppresses the Arc--gene transcription responding to the tone--shock paired stimuli in the LA neurons." This statement is incorrect as well. See comment 2 above.

24. "(1) the Ldb2 protein in LA neurons suppresses Arc-gene expression during fear conditioning," is incorrect. It should be that lack of LDB2 protein in LA neurons increases Arc expression during fear conditioning.

25. Is there any functional differences between cells in LA and BAI? Arc expression level in BAI is same between WT and KO. Are Arc expressing neurons in BAT not contributing to fear learning?

26. How do you interpret the increased LA Arc expression in unpaired group of KO? What do you think is initiating Arc expression in unpaired and paired group of WT and KO? This needs explanation.

27. "(2) the Ldb2 KO mice fail to induce appropriate plasticity of synapses due to Arc overexpression, consequently leading to impaired tone-related fear conditioning ability" this is not correct conclusion. The patch clamp experiment (looking at synaptic activity) didn't show any differences between WT and KO although there was Arc expression differences between WT and KO. The authors didn't do any experiment investigating synaptic plasticity and the mEPSC measured by patch clamp is not measuring synaptic plasticity. The conclusion drawn here and the following hypothesis (Fig7) is not correct hence needs modification.

28. In Fig7, there are "miniature EPSP ↓" & "Fear Learning ↓". This is not reflecting their findings as KO mice DID NOT show reduced mEPSCs but DID show reduced fear learning.

Fourth result section: Neurogenesis is reduced in the hippocampus of Ldb2 KO mice

29. Supple Fig 6 legend has an error in animal numbers. Correct "three animals for each genotype" to six animals for each genotype. Also report how many sections were quantified per animal. Representative pictures should also be included.

Fifth result section: Ldb2 forms complexes with known transcription-related proteins in vivo

30. The section title is wrong. The interaction was not measured in vivo. It was done in vitro. This needs correction.

31. In Fig8, there are green and black arrows, explain what they are in the legend.

Sixth result section: ChIP-seq analysis

32. Add brief explanation about each cell lines used such as ECC1, H1ES, HCT116 and K562.

33. For the GO analysis, what was the background gene set? Use of correct background gene set is important as this will affect the enrichment result and interpretation of the result. There is no information on how GO analysis was performed in the method/supple method section.

34. What is the explanation of other GO terms listed in Table 2 for the function of LDB2 and EGR?
35. "disruption of the 'LDB2-EGR axis' has a role in the etiology of mental disorders" The results shown in the study does not directly support the etiology of mental disorders. The LDB2-EGR axis is a possible mechanism under the phenotypes seen in LDB2 KO mice, NOT mental disorders. This needs to be made clear.

Discussion

36. "LDB and LDB2-mediated gene expression networks have an important role in the pathogenesis of schizophrenia and other mental disorders, including bipolar disorder" The current study did not show any direct evidence of this, hence should be modified to accurately describe their findings.

37. Please explain what "the face and construct validities" in this context. It is not clearly explained.

38. "We also presented evidence that Arc/ARC-mediated synaptogenesis is one of downstream targets for LDB2/Ldb2 in the amygdala," authors did not show any data on synaptogenesis. This was their guess. This needs to be made clear.

39. "Our study revealed that the Ldb2 deficiency results in an excess of the Arc expression in excitatory neurons in the amygdala, probably causing the suppression of mEPSCs" Ldb2 KO mice had same frequency and amplitude of mEPSCs (Supple Fig5), therefore, the increased Arc expression is not affecting mEPSCs at all. This needs correction. In addition, the increased Arc expression in KO compared to WT was only observed in LA not BAI in amygdala. This needs to be made clear.

40. One of the significant finding was reduced neurogenesis in DG in hippocampus hence it needs to be discussed. Also explain how neurogenesis could have been affected in KO when LDB2 is not expressed in DG in WT mice.

Response to the Reviewers

Response to Referee #1:

Review of Cooperation of LIM domain binding 2 with EGR transcription factors to regulate gene expression networks relevant to mental disorders. Ohnishi et al studied a role of LIM domain binding 2 (LDB2) gene encoding a putative transcription regulator in mental disorders. In the present study, they used different approaches, phenotypic analyses of *Ldb2* mouse model; yeast two-hybrid, identification of direct genomic targets of *Ldb2* by chromatin immunoprecipitation (IP) followed by sequencing (ChIP-seq). The present manuscript deserves publication with minor revisions.

Our response:

We would like to thank you for your reading our manuscript and supporting publication of it in EMBO Molecular Medicine. We have addressed all the your comments as follows:

1. Mouse model data: pp 9-15 and Figures 2-6.

For non-specialist reader, I think that it can be useful to resume phenotypical traits that characterize bona fide mouse models of schizophrenia. These phenotypical traits include neuronal architecture, synaptic plasticity, circuitry deficits and behavioral paradigms dependent on learning such as spatial working memory (WM) performance using a T-maze delayed non-match to place (DNMTP) task; Pre-Pulse Inhibition (PPI) (see for instance Gogos et al., 2019; Mukai et al., 2019). This information can highlight translational phenotypes from mouse to humans.

Our response:

According to the suggestion, we have added some behavioral tests and summarized them in Supplementary Table (Appendix Table S1). In particular, we conducted the 8-arm radial maze and Barnes maze tests for the working memory assessment, in addition to Y-maze test. However, no signs of working memory deficits in the KO animal were detected. We also evaluated the social behavioral function of KO mice using the encounter method, but again no signs of social behavioral deficit were seen. Please pay attention to that we only claimed that *Ldb2* KO mice 'at least partly' fulfill the behavioral or neuronal criteria as a schizophrenia or bipolar model. Although we agree that investigating the deficits in neuronal architecture in KO mice are critical, we believe that it should be assessed in the future study.

We have referred to Appendix Table S1 in the "Ldb2 KO mice display pleiotropic behavioral abnormalities related to mental disorders" section, as follows:

Revised) To explore the possibility that dysregulated expression of *LDB2* had an impact on the manifestation of schizophrenia, we next set out to analyze phenotypes of *Ldb2*-deficient mice (Fig 2, Fig EV3, Appendix Table S1) employing our behavioral test batteries.

2. Pages 11-14: Analysis of Arc expression as a function of a cognitive test. The authors generated Arc Venus tg/tg; Ldb2 KO (AV KO), and Arc Venus tg/tg; Ldb2 WT (AV WT) to monitor neuronal activities in the amygdaloid complex through the Venus fluorescence. They found that the increment of the Middle Venus positive neurons was significantly larger in the AV KO compared to AV WT. These results indicating that Ldb2 suppresses the Arc gene transcription responding to the tone shock paired stimuli in the LA neurons (Figure 5). Synaptic activity is reduced as a function of Arc expression whatever the genotype (Figure 6 and text page 14). The logic is to present the result (Figure 6) before the analysis of the number of positive neurons (Figure 5), in order to present first the relation between arc expression and synaptic activity and second the inverse relationship between the number of ARC expression and the level of Ldb2 protein. Furthermore, I think that indication of regression in graphs of Figure 6 with a graph for each condition will increase the readability of the figure.

Our response:

Thank you for the constructive comment. We have moved the original Fig. 6 (now Fig. 5) ahead of the original Fig. 5 (now Fig. 6) as suggested by this reviewer. Also, we have added the regression lines in new Fig. 5 for the readability.

3. Discussion

Phenotypic traits of the mouse model need to be discussed in comparison to what found in bona fide mouse models.

Our response:

Considering the complexity of the pathogenic mechanism of schizophrenia, we think it difficult to say which are the *bona fide* mouse models for the illness. However, as explained in the Introduction section, *DISC1* is an established causal gene. Therefore, we have added statements where the behavioral phenotypes of *Ldb2* KO mice were compared to those of *Disc1*-modified animals, to the Discussion section as follows:

(Discussion)

Original) Second, ablation of *Ldb2* in a mouse resulted in pleiotropic behavioral abnormalities reminiscent of schizophrenia and bipolar mania. Moreover, some of behavioral alternations were alleviated by antipsychotics or a mood stabilizer (lithium), suggesting that the *Ldb2* KO animal satisfies the predictive validity as a model for schizophrenia and bipolar mania, as well as the face and construct validities.

Revised) Second, ablation of *Ldb2* in a mouse resulted in pleiotropic behavioral abnormalities reminiscent of schizophrenia and bipolar mania. Those phenotypes include hyper-activity in home cage or open field, reduced alternation in the Y-maze test, severe deficits in fear conditioning, and exaggerated responses to methylphenidate or MK-801. Similar phenotypes have been repeatedly reported in many of animal models for schizophrenia or bipolar disorder including *Disc1* gene modified mice (Ref). Moreover, some of behavioral alternations were alleviated by antipsychotics or a mood stabilizer (lithium), suggesting that the *Ldb2* KO animal satisfies the predictive validity, which is justified by expected responses of the model to therapeutics including medication, as a model for schizophrenia and bipolar mania. The animal also satisfies the face and construct validities, since its phenotypes recapitulated some symptoms of the diseases and the phenotypes were elicited by the disruption of the gene, whose homologue is located in the vicinity of the chromosomal breakpoint seen in the proband. However, *Ldb2* KO mice did not show clear deficits in prepulse inhibition, which has been considered one of endophenotypes for mental disorders such as schizophrenia.

4. Results of Figure 6 have to be discussed with the indication that the decrease in mEPSPs possibly reflect a change in the number of postsynaptic AMPA receptors.

Our response:

Thank you for the valuable comment. We have included the statement concerning the relationship between Arc and postsynaptic AMPA-receptor expression (P12, L).

Small corrections:

Figure 1A Dimerization instead of Dimerization

Our response:

We have corrected the typo.

Response to Referee #2:

The authors present an extensive modeling study using both mouse and iPSC-derived neurospheres to examine the potential for a novel risk gene for schizophrenia. An alternative cellular model would be iPSC-derived neurons as well as iPSC-derived neural cells from the patient.

Our response:

Thank you for the comment. Ldb2 expression is prominent in neurospheres. Neurospheres are enriched in neural progenitors. As for iPSC-derived neurons, it was very difficult for us to ensure sufficient cell number for ChIP analysis.

Comments

1. This manuscript reports on a tour-de-force effort to use both animal and human cellular models to examine various functional aspects of LDB2, a gene located close to the breakpoint on 4p16, a part of a very rare chromosomal structural aberration in a schizophrenia patient. The findings presented are important and additional supportive future data would heighten the relevance of the LDB2-EGR system to risk for schizophrenia.

Our response:

We would like to express our thanks to the positive comments.

Minor comments

2. If the parents of the proband were available, were they unaffected and did any of them carry the translocation?

Our response:

The proband's parents were unaffected and did not carry the chromosomal abnormality, supporting the idea that the translocation was causal in the proband. Please find that these observations are explained in the Introduction section.

3. Are there any known GWAS hits on 4p16.1 & 13q21.31 for any neuropsychiatric disorder?

Our response:

A large GWAS of schizophrenia (Nature 511, 421-, 2014), did not detect any signals located at chr. 13. Although 4 loci were detected in chr. 4, even the nearest one was > 16 Mbp distant from the breakpoint. A GWAS of bipolar disorder, which was published last year (Stahl et al., 2019, Nat. Genetics), identified only one locus in chr. 4. The SNP with the strongest *P* value (located within the

FSTL5 gene) in the locus was > 7 Mbp distant from the breakpoint. No chr. 13 signals were detected in the study. These observations favor to the idea that common variants in the *LDB2* locus are not likely to contribute to the genetic susceptibility of schizophrenia or bipolar disorder.

For readers' convenience, we have added a statement to briefly explain these observations to the last part of 2nd paragraph of the Discussion section as follows:

Revised) However, it should be noted that prior GAWAS has not supported that common variants in the *LDB2* locus contribute to the genetic susceptibility of schizophrenia or bipolar disorder.

4. Did the schizophrenia genomic DNA carry any of the known pathogenic CNVs, e.g., on 15q, 16p or 22q?

Our response:

Thank you for raising this critical issue. The NGS analysis has shown that the proband's genome harbors large deletions in two chromosomes: chromosome 6 (10,657,982-10,660,876; 2,885 bp) and chromosome X (55,702,373-55,709,904; 7,447 bp) as shown below.

Due to the unavailability of genomic DNAs from the proband's parents, we were not able to determine whether these deletions were *de novo* or inherited from one of the proband's parents, who manifested no psychiatric symptoms. Although this makes it difficult to evaluate the pathogenic role of these deletions in the proband, no known genes are mapped to these two deleted regions. Moreover, to our best knowledge, no reports have identified these regions as susceptibility loci for schizophrenia. Collectively, these observations do not favor to the idea these two deletions were pathogenic in the proband. We have reflected these observations in the Introduction section (2nd paragraph) as follows:

By reanalyzing the NGS data (Horiuchi et al.) carefully, we found that the proband's genome harbors large deletions in two chromosomes: chromosome 6 (10,657,982-10,660,876; 2,885 bp) and chromosome X (55,702,373-55,709,904; 7,447 bp). Due to unavailability of genomic DNAs from the proband's patients, we were not able to determine whether these deletions were *de novo* or inherited from one of the proband's parents, who manifested no psychiatric symptoms. However, no known genes are mapped to these two deleted regions. Moreover, to our best knowledge, no reports have identified these regions as susceptibility loci for schizophrenia or bipolar disorder.

5. How far is LDB2 from the 4p breakpoint?

Our response:

32.6 kbp. To clarify this point, we have modified the Introduction section (second paragraph) as follows:

Revised) The breakpoint on chromosome 4 is mapped to the **32.6-kbp** upstream region of a gene encoding a putative transcription regulator lacking a DNA-binding domain, LIM domain-binding 2 (*LDB2*), also called *CLIMI* (Horiuchi et al. Human Genome Variation, 2020) (Supplementary Fig. 1).

6. Did the authors try to sequence the 2 bands on Western blot?

Our response:

Although we have not conducted amino acid sequencing of the two proteins, our monoclonal and polyclonal antibodies, both of which were directed against the N-terminal portion of mouse *Ldb2*, commonly reacted to the two proteins. In addition, the two bands disappeared in western blot when brain lysate from a KO animal was used. Based on these observations, we concluded that the two bands represent the *Ldb2* gene products.

7. Why did the authors use iPSC-derived neurospheres instead of either iPSC-derived neural progenitor cells or neurons?

Our response:

The reason is that *Ldb2* expression is prominent in neurospheres. Neurospheres are enriched in neural progenitors. As for iPSC-derived neurons, it was very difficult for us to ensure sufficient cell number for ChIP analysis.

8. It would have been interesting also to use iPSC-derived neural cells from the patient.

Our response:

We agree with the comment. We would like to analyze cellular phenotypes of the proband iPSC-derived neural cells in the future study. In the 1st paragraph of Discussion section, we have mentioned this as follows:

Revised) In the future study, the cellular phenotypes of these cells should be addressed.

9. Y-2-H tends to have problems with signal-to-noise ratio, any comments about this issue?

Our response:

As the reviewer suggested, Y-2-H sometimes produces substantial noise signals. As shown in the original version, we confirmed the protein-protein interactions using coimmunoprecipitation assay. To clarify this point, we have amended the Result section as follows:

Original)translin-associated factor X (TSNAX), protein kinase, muscle-type (PKM2), and CD9 and/or sulfide quinone oxidoreductase (SQOR), were also identified (**Table 1**). Next, we performed co-immunoprecipitation assay by focusing on the proteins that potentially act as transcription factors or transcription regulatory factors.

Revised)translin-associated factor X (TSNAX), protein kinase, muscle-type (PKM2), and CD9 and/or sulfide quinone oxidoreductase (SQOR), were also identified (**Table 1**). **Since two-hybrid screening is known to sometimes produce false-positive clones, we validated the specificities of the potential interactions with** co-immunoprecipitation assay, especially focusing on the proteins that potentially act as transcription factors or transcription regulatory factors.

10. There are too many numbers in pages 12-14 that are distracting. Presenting these numbers in Tables would help.

Our response:

According to this comment, we have moved some of numbers to figure legends.

11. Fig. 1A: Replace "dimerization" with "dimerization"

Our response:

We have corrected it.

12. Fig. 1D, F & G: An improvement in color intensity would help.

Our response:

We have created new Fig.1 with improved color contrast.

13. Fig. 2: Indicate in the legend what blue and red lines represent.

Our response:

We have added information required to the legends.

14. Fig. 4A: Replace "venral" with "ventral"

Our response:

We have corrected it in new Fig. 6.

15. Figs. 5 & 6: Revise the title to indicate what the figures are showing in a manner similar to how the title in Fig. 4 is constructed.

Our response:

We have modified them as follows:

Fig. 5: The amplitude and frequency of miniature EPSCs of Venus-positive cells are inversely correlated with the fluorescence intensity

Fig. 6: The number of Venus-positive neurons in the lateral amygdala is larger in fear-conditioned KO mice

16. Fig. 7: Include a legend.

Our response:

We have decided to remove this figure because we found no significant difference in mEPSC parameters.

17. Fig. 8: The legend can be made more descriptive. Fig. 8F legend is missing.

Our response:

We have amended it as follows:

Results of coimmunoprecipitation between overexpressed LDB2 and interaction candidates are shown (A-D, F). **LDB2 vs. LDB2 (A), LDB2 vs. LHX2 (B), LDB2 vs. LMO2/3/4 (C), LDB2 vs. SSBP2/3/4 (D), and LDB2 v.s RLIM (F).** LDB2 (WT, T83N or P170L) and candidate interactors were tagged with Flag and HA, respectively. **E.** Coimmunoprecipitation of endogenous Ssbp3 and Lmo3 with Ldb2. Immunocomplexes prepared **from mouse brain lysate** using an antibody against Ldb2, Lmo3 **or** Ssbp3, or a control antibody were analyzed by western blotting using an anti-Ldb2 (left) or anti-Ssbp3 (**right**) antibody.

18. Fig. 10: The model can be simplified. You may need to present just the top figure, revised to reflect both activation & repression

Our response:

Accroding to the suggestion, we have simplified the Figure as follows:

Figure10

Response to Referee #3:

Use of KO mice to study the function of the gene (LDB2) is appropriate. However, to insist that the model is a mental disorder model is not appropriate. They need to modify their statement in the manuscript.

Our response:

Thank you for the comments. We have amended some parts of the manuscript as explained in the later parts of our response.

Ohnishi et al. have investigated the expression pattern of Ldb2 in mice, effect of Ldb2 KO in mouse behaviours and Arc expression and identified direct interactors of LDB2 and potential indirect binding sites in DNA. They have postulated possible mechanisms of phenotypes seen in the KO mice and the action of LDB2. This is the first study generating and studying Ldb2 KO mouse and in that sense, this study is novel and provides data that could benefit others. However, the link between the schizophrenia and the gene is very weak as it is based on one case although de novo, hence linking the findings of this study to mental disorders, as was stated in several places throughout the manuscript by the authors, is not appropriate and over-stretching.

Another point that has to be corrected is the interpretation of their data and the building of hypothesis and model. The manuscript shows either no supporting evidence or rather opposite evidence to build their hypothesis, especially around Arc expression and mEPSCs. Also it has to be noted that mEPSC recording is not a measure of synaptic plasticity in this experimental setting. Often the statements were not directly supported by the results, therefore, either new experimental data needs to be added to support their statements or they need to be modified to reflect the data.

Our response: Thank you for the insightful comments. We have rewritten some parts of the manuscript as explained in the later parts of our response. As for the analysis of mEPSCs and its interpretation, we have rewritten the manuscript thoroughly.

Introduction

1. Do patient derived tissues (any somatic tissue? Biopsy?) show reduced expression of LDB2? In order to support the breakpoint is affecting the expression of LDB2, this data is needed.

Our response:

Thank you for pointing out the critical point. Although tissue samples from the patient were not available, we have stored the EBV-transformed lymphoblastoid cells from the patient and a healthy control (ID: HAM-C-180, 39 y.o, male). Since the *LDB2* transcripts were not detectable in the cells,

we have decided to newly establish iPS (LiPS) cell lines each from the patient and the healthy control subject and differentiated them into neurospheres. We have attached the data of qRT-PCR (below).

In both cell conditions, we detected significantly lower expression of *LDB2* in the patient than in control. We have briefly mentioned these results in the first paragraph of the result section as follows:

Revised) These observations support that the chromosomal translocation affected the expression of the *LDB2* transcript.

Consistent with this idea, we detected a lower expression in LiPS cells (established from lymphoblastoid cells) and LiPS cell-derived neurospheres from the patient than a healthy control (39 y.o, male) We present these data as new Fig. EV2.

Fig EV2

2. Has the lab-made LDB2 antibody tested for cross reactivity for LDB1?

Our response:

We have carefully designed the antigen to avoid crossreactivity of the antibody to Ldb1/LDB1 (Fig. 1A, B). In fact, the antigen region was the most divergent between the two proteins. In addition, we have experimentally confirmed the specificity of our polyclonal antibody. There were essentially no bands in the brain tissue lysate from *Ldb2* KO mice (Fig. EV3C), supporting that the polyclonal antibody is specific to Ldb2. As for our monoconal antibody, we have confirmed the specificity using tissue sections form *Ldb2* KO mice (Fig. EV3F).

3. Fig 1D doesn't necessarily show that Ldb2 is expressed only in neuron. Remove referring of Fig.1D in the first result section claiming this.

4. Fig 1D doesn't show that layer V has most prominent expression. Layer II/III also has strong expression too. Reflect this in the result. Proper quantification is needed with together with NeuN staining in all layers.

5. Fig 1E doesn't prove that Ldb2 expression is highest in layer V. Do not refer Fig 1E when claiming this.

6. When claiming in the expression in Amygdala, there is no evidence showing the expression is only in neurons. Should do co-staining with NeuN to claim this.

Our response to #3-6:

We have conducted additional experiments and reflected the results to new Fig. 1 as follows:

Figure 1

These data clearly show that > 80% of Ldb2-positive are NeuN-positive. In addition, we have presented the staining pattern of Ldb2 in the cerebral cortex, hippocampus and amygdala from KO mice, evidencing the specificity of the antibody we have created (Fig EV3F).

Fig. EV3

We have amended the result section to explain these data (second result section: *Ldb2* KO mice display pleiotropic behavioral abnormalities relevant to mental disorders).

7. In the first line ‘dysregulated expression of LDB2’ should be lack of LDB2 as the model they use is deficiency rather than dysregulation.

Our response:

We have removed ‘To explore the possibility that dysregulated expression of *LDB2* had an impact on the manifestation of schizophrenia,’ from the sentence.

8. "Although *Ldb2* showed a layer-specific expression in the cerebral cortex," This was not investigated fully. The authors need to quantify the % of neurons expressing LDB2 in each cortical

layer to support this. Even in the representative picture in Fig 1D, it doesn't seem it has specific expression in layer V.

Our response:

Please refer to our response to #3-6 and new Fig. 1.

9. In Fig. 2, write the number of mice used in each test either in the figure itself or in the figure legend. Same goes for the rest of figures.

Our response:

We have added the information.

10. Details of the behavioural test protocol needs to be written in the supplementary information or method.

Our response:

We have added information on the protocols for behavioral study in Supplementary Information.

11. Add statistical results in Fig 3A and B bar graphs in comparing WT/saline vs WT/MPD and WT/saline vs WT/MK-801.

Our response:

Those were not significant.

12. Fig3A bar graph Y axis label is wrong.

Our response:

We have corrected it.

13. Explain why the dose of MK-801 went up from 0.2 mg/kg in Fig 2B locomotion test into 0.5 mg/kg in 'popping' behaviour test. Did 0.2 mg/kg MK-801 not produce statistically significant result in popping behaviour? If so, this needs to be mentioned.

Our response:

Although we have conducted preliminary experiment in lower doses using a small number of animals, the effect of the drug was relatively weak, when compared to 0.5 mg/kg.

14. Looking at Fig 2B open field test, KO mice are hyperactive. However, in Fig 3A and B, the locomotion between KO and WT before the injection (effectively same as Fig 2B open field test

then) does not show hyperactivity even in the first 10 mins. This raises a question if the increased locomotion in open field test (Fig 2B) was by chance rather than consistent observation.

Our response:

In the home cage test, the activities in the day 1 were not different between KO and WT (Fig. 2A). However, the activities in the day 2 and thereafter were higher in KO than WT (Fig. 2A). Therefore, we believe that the KO mice intrinsically have a hyperactive trait and the data presented in Fig. 2B was not by chance.

15. Fig. 4A Haloperidol graph has unnecessary stars above WT saline. Remove them.

Our response:

We have removed the haloperidol data. Please refer to our response to #17.

16. Was the locomotion between WT and KO after saline injection significantly different? There is no significance recorded in the figure so it is assumed that were not different. Then again as commented above (14), the increased locomotion in open field test (Fig 2B) was by chance rather than consistent observation.

Our response:

In the open field test (Fig. 2B), hyperlocomotion was evident during the first 5 min. In contrast, cumulative locomotor activities in home cage in the day 1 were not different between KO and WT (Fig. 4A). Therefore, the hyperlocomotion of the KO mice may be statistically masked when the locomotor activity is measured during more than 5 min in the novel place. We agree with the reviewer, in that a hyperlocomotive trait of KO mice is evident in some conditions. But the detailed mechanism remains elusive.

17. "Importantly, deficits in fear-conditioning test as well as the hyperlocomotive trait of the KO mice were ameliorated by the treatment with typical (haloperidol) and atypical (clozapine) antipsychotics (Fig. 4A, B)." This sentence is not correct. Fig 4B Haloperidol result doesn't show significant improvement in KO after haloperidol injection. Reflect this in the text. In addition, in both open field and fear conditioning test, both the WT and KO responded to drugs. Therefore, authors cannot justify the suitability of LDB2 KO as schizophrenia model based on this. If so, even the WT would've been considered as the disease model as they have responded to the drug.

Our response:

We were not able to determine the drug dose in which biological effects are seen in only KO animals. We have removed Fig. 4A and the haloperidol data from Fig. 4B, and revised the related description.

18. Add individual mouse data into the bar graphs in Fig 4 as was done in other figures (Fig 2, 3).

Our response:

We have modified the graphs.

19. Fig2A shows significantly increased activity of KO mice in home case both in dark and light phases. In figure 4C, KO has higher activity during dark phase in no lithium carbonate group; however, in the light phase there is no activity difference between WT and KO without lithium carbonate. How do you explain this? Why KO show no hyperactivity in Fig4C in light phase whereas there is a significant hyperactivity in Fig2A in light phase?

Our response:

Please be aware that the basal activity in the light phase is much lesser than in the dark phase. In our opinion, this sometimes makes it difficult to reproducibly detect relatively small differences.

20. For contextual test, there isn't haloperidol result. Why is that? Was it not performed or was it not significant?

Our response: It was not performed. We have removed all haloperidol result in the revised manuscript.

21. How many coronal sections were used for counting venus+ cells? Please specify.

Our response:

Thank you for the comment. We have now added the information in the text.

22. "that Ldb2 of excitatory neurons suppressed the number of the Arc gene-expressing neurons in response to the tone-shock paired or unpaired stimuli in the LA." This is not correct. Fig5B LA graph shows a significant increase of Venus+ cells (Arc expressing cells) in WT paired group. This means that Ldb2 in excitatory neurons did not suppress the expression Arc. If that was the case, then the WT paired group would show no increased in venus + cells in comparison to naïve and unpaired WT groups. The conclusion from Fig5B LA graph needs re-writing to accurately reflect the result.

Suggestion: Lack of *Ldb2* expression in excitatory neurons in LA of amygdala promoted the expression of *Arc* upon stimuli during fear conditioning trials.

Our response:

Thank you for the comment. We have rewritten the text to reflect the result accurately.

23. "These results indicate that *Ldb2* suppresses the *Arc*—gene transcription responding to the tone--shock paired stimuli in the LA neurons." This statement is incorrect as well. See comment 22 above.

Our response:

Thank you for the comment. We have corrected it.

24. "(1) the *Ldb2* protein in LA neurons suppresses *Arc*-gene expression during fear conditioning," is incorrect. It should be that lack of *LDB2* protein in LA neurons increases *Arc* expression during fear conditioning.

Our response:

We have rewritten the conclusion.

25. Is there any functional differences between cells in LA and BAI? *Arc* expression level in BAI is same between WT and KO. Are *Arc* expressing neurons in BAT not contributing to fear learning?

Our response:

Thank you for the valuable comment. The LA receives many sensory inputs directly from the thalamus or indirectly through the cortex, while the BAI receives inputs mainly from the LA. Cued-fear learning, in which severe deficits were observed in the *Ldb2* KO mice, is thought to depend on synaptic plasticity in LA neurons. On the other hand, contextual-fear learning does not depend on the LA. In fact, the increase of *Arc*-gene expression was observed only in the LA of fear-conditioned mice in the *Arc*-Venus analysis. Meanwhile, *Arc*-gene expression in the BAI was indistinguishable between genotypes. These results suggest that the change in *Arc*-gene expression in the BAI is not associated with cued-fear learning, and that the BAI is unlikely to contribute to cued-fear learning significantly.

26. How do you interpret the increased LA *Arc* expression in unpaired group of KO? What do you think is initiating *Arc* expression in unpaired and paired group of WT and KO? This needs explanation.

Our response:

Thank you for the constructive comment. The reason why Arc expression was increased in the unpaired group is unclear in the present study. One possibility is that, in the LA, excitability of excitatory neurons is lowered by Ldb2 and neurons will not fire unless the input of the foot shock (pain) and tone comes together. However, when the inputs of both pain and tone converge simultaneously to the neurons, they will fire and LTP occurs in those neurons, and subsequently, only the input of tone causes freezing behavior (cued-fear learning). On the other hand, the inhibitory mechanism is weakened in KO mice and the excitability of neurons is heightened nonspecifically, which makes it difficult to distinguish the sensory inputs and to establish the fear learning. In the BA, however, the inhibitory mechanism is weak from the beginning and neurons will fire nonspecifically in response to the input of tone or pain. Because this possibility is not supported by experimental results and, in that sense, too speculative, we do not describe it in the Discussion section. However, this issue is very important and we would like to pursue it in the future study.

27. "(2) the Ldb2 KO mice fail to induce appropriate plasticity of synapses due to Arc overexpression, consequently leading to impaired tone-related fear conditioning ability" this is not correct conclusion. The patch clamp experiment (looking at synaptic activity) didn't show any differences between WT and KO although there was Arc expression differences between WT and KO. The authors didn't do any experiment investigating synaptic plasticity and the mEPSC measured by patch clamp is not measuring synaptic plasticity. The conclusion drawn here and the following hypothesis (Fig7) is not correct hence needs modification.

Our response:

Thank you for the comment. We agree with the reviewer and have rewritten the conclusion completely. Although there was no difference in mEPSC parameters between genotypes, the neuron that expresses more Arc shows smaller median amplitudes and lower frequency of mEPSCs regardless of genotypes. Therefore, it can safely be concluded that "(2) the amount of cellular Arc expression is inversely correlated with spontaneous synaptic activity in the LA".

28. In Fig7, there are "miniature EPSP ↓" & "Fear Learning ↓". This is not reflecting their findings as KO mice DID NOT show reduced mEPSCs but DID show reduced fear learning.

Our response:

Thank you for the comment. We have deleted Figure 7 because there was no difference in mEPSC parameters between genotypes and because this figure is somewhat speculative.

Fourth result section: Neurogenesis is reduced in the hippocampus of Ldb2 KO mice

29. Supple Fig 6 legend has an error in animal numbers. Correct "three animals for each genotype" to six animals for each genotype. Also report how many sections were quantified per animal. Representative pictures should also be included.

Our response:

As the reviewer pointed out, we have used 3 animals/genotype, and both hemispheres were analyzed, resulting in $n = 6$ (2×3). We believe that many studies including ours have conducted this type of replication when brain tissue sections were analyzed. Unfortunately, we have not stored the photographs, since we have counted the positive cells by real-time observation.

Fifth result section: Ldb2 forms complexes with known transcription-related proteins in vivo

30. The section title is wrong. The interaction was not measured in vivo. It was done in vitro. This needs correction.

Our response:

We have removed '*in vivo*'.

31. In Fig8, there are green and black arrows, explain what they are in the legend.

Our response:

We have added information as follows:

Revised) Green and black arrowhead indicate positions for Flag-Ldb2 and HA-proteins, respectively.

Sixth result section: ChIP-seq analysis

32. Add brief explanation about each cell lines used such as ECC1, H1ES, HCT116 and K562.

Our response:

We have explained the cell lines in the sixth Result section, and we have added a following sentence to the Method section:

ChIP-seq data of the EGR1 were obtained using cancer-derived cell lines, ECC1 (endometrial cancer), HCT116 (colon cancer) and K562 (erythromyeloblastoid leukemia), and using an ES cell line H1ES from the NCBI SRR.

33. For the GO analysis, what was the background gene set? Use of correct background gene set is

important as this will affect the enrichment result and interpretation of the result. There is no information on how GO analysis was performed in the method/supple method section.

Our response:

We are sorry for the confusion. As explained in the text, Appendix Table S1 lists gene names extracted from the common ChIP peaks between LDB2 and EGR1 in the ECC1 and K562 lines. The GO analysis was done using this gene set by comparing to all genes annotated. We have modified the description in the result section as follows:

Original) in the ECC1 and K562 lines were enriched in the upstream regions or 5' UTRs of target genes (**Fig. 9E**). A Gene Ontology analysis (<http://geneontology.org>) using the set of genes with overlapping ChIP-seq signals between the LDB2 and EGR1 (Supplementary **Table 1**),

Revised) in the ECC1 and K562 lines were enriched in the upstream regions or 5' UTRs of target genes (**Fig. 8E, Appendix Table S2**). A Gene Ontology analysis (<http://geneontology.org>) using the genes listed in **Appendix Table S1**.

34. What is the explanation of other GO terms listed in Table 2 for the function of LDB2 and EGR?

Our response:

Currently, we have no reliable information on how other biological processes can be related to the phenotypes of *Ldb2* KO mice in terms of the pathophysiology of schizophrenia.

35. "disruption of the 'LDB2-EGR axis' has a role in the etiology of mental disorders" The results shown in the study does not directly support the etiology of mental disorders. The LDB2-EGR axis is a possible mechanism under the phenotypes seen in *Ldb2* KO mice, NOT mental disorders. This needs to be made clear.

Our response:

We have weakened our claim as follows:

Revised) we speculate that LDB2/*Ldb2* cooperates with the EGR family via unknown protein(s) to control expressions of numerous genes including *ARC* (**Fig. 10**), and that 'LDB2-EGR axis' may potentially underlie the etiology of subset of mental disorders.

Discussion

36. "LDB and LDB2-mediated gene expression networks have an important role in the pathogenesis of schizophrenia and other mental disorders, including bipolar disorder" **The current study did not show any direct evidence of this, hence should be modified to accurately describe their findings.**

Our response:

We have modified the sentence as follows:

Original) The current study provided multiple lines of experimental evidence that *LDB* and *LDB2*-mediated gene expression networks have an important role in the pathogenesis of schizophrenia and other mental disorders, including bipolar disorder.

Revised) Multiple observations in the current study favor to the idea that *LDB* and *LDB2*-mediated gene expression networks play a potential role in the pathogenesis of subset of schizophrenia and other mental disorders, including bipolar disorder.

37. Please explain what "the face and construct validities" in this context. It is not clearly explained.

Our response:

We have explained them as follows:

face validity: a similarity exists between behavioral phenotypes of a model animal and disease symptoms

construct validity: a model animal can be generated based on a mechanistic theory of disease

38. "We also presented evidence that Arc/ARC-mediated synaptogenesis is one of downstream targets for *LDB2/Ldb2* in the amygdala," authors did not show any data on synaptogenesis. This was their guess. This needs to be made clear.

Our response:

We have change the text as follows:

Original) We also presented evidence that Arc/ARC-mediated synaptogenesis is one of downstream targets for *LDB2/Ldb2* in the amygdala,

Revised) Our ChIP-seq analyses suggested that binding sites of *LDB2* and *EGR* proteins in human cells are concentrated to synaptogenesis-related genes including *ARC*. The expression of *Arc* was found to be modulated in the amygdala of *Ldb2* KO mice. Therefore, it is tempting to speculate that synaptogenesis is one of the downstream targets for *LDB2/Ldb2* in the amygdala,

39. "Our study revealed that the *Ldb2* deficiency results in an excess of the Arc expression in excitatory neurons in the amygdala, probably causing the suppression of mEPSCs" *Ldb2* KO mice had same frequency and amplitude of mEPSCs (Supple Fig5), therefore, the increased Arc expression is not affecting mEPSCs at all. This needs correction. In addition, the increased Arc

expression in KO compared to WT was only observed in LA not BAI in amygdala. This needs to be made clear.

Our response:

Thank you for the comment. We agree with the reviewer. We have rewritten the statement as follows:

“Our study revealed that the amount of cellular Arc expression is negatively correlated with synaptic activity in the LA, and that Ldb2 deficiency results in an excess of the Arc expression only in the LA, which probably causes the disturbance of auditory fear learning”.

- We have now included the words “only in the LA” to emphasize that the increase of Arc expression was restricted to the LA.

40. One of the significant finding was reduced neurogenesis in DG in hippocampus hence it needs to be discussed. Also explain how neurogenesis could have been affected in KO when LDB2 is not expressed in DG in WT mice.

Our response:

Thank you for raising this critical point. We have currently no idea on the mechanism of reduced neurogenesis. For the readers’ understanding, we have restructured the result section, pointing out that no clear expression of the Ldb2 protein was seen in the DG as follows:

Revised) Neurogenesis in the hippocampus has a role in this learning. In addition, impairments in adult neurogenesis have been repeatedly reported in schizophrenia, affective disorders, and their model animals. We found that the neurogenesis in hippocampal dentate gyrus (DG) was remarkably decreased in the KO mice, although no clear expression of the Ldb2 protein was seen in the DG.

4th Jan 2021

Dear Dr. Ohnishi,

Thank you for the submission of your revised manuscript to EMBO Molecular Medicine. We have now received the enclosed report from the three referees who were asked to re-assess it. As you will see the referees are now overall supportive and I am pleased to inform you that we will be able to accept your manuscript pending the following amendments:

1. please address the minor concerns raised by the referees.
2. In the main manuscript file, please do the following.

***** Reviewer's comments *****

Referee #1 (Remarks for Author):

Cooperation of LIM domain-binding 2 (LDB2) with EGR in the pathophysiology of schizophrenia

The authors extensively analyzed Ldb2 knockout (KO) mice that displayed multiple deficits relevant to mental disorders.

In particular, they found deficits in the fear-conditioning paradigm, a possible dysregulation of synaptic plasticity controlled by the immediate-early gene ARC is involved in the phenotypes. LDB2 forms protein complexes with known transcription factors. Consistently, ChIP-seq analyses. They completed this mouse model analysis with human neurosphere analysis. They found that LDB2 binds to >10,000 genomic sites, with many of those sites, including the promoter region of ARC, are occupied by EGR

Altogether these results suggest that dysregulation in the gene expression controlled by the LDB2-EGR axis underlies a pathogenesis of subset of mental disorders.

As indicated by the authors in the revised manuscript "Collectively, the Ldb2 KO mice at least partly fulfill the criteria of both face validity (a similarity exists between behavioral phenotypes of a model animal and disease symptoms) and predictive validity (a drug used for a disease is also effective to ameliorate phenotypes of a model animal) for schizophrenia and bipolar mania".

The revised manuscript takes in account the comments of the three reviewers.

Novel experimental results have been added:

- (i) New behavioral analysis are now summarized in Appendix Table S1. Summary of behavioral analyses of Ldb2 KO mouse.
- (ii) Validation of Yeast Two-Hybrid results by immunoprecipitation (Fig.7E)
- (iii) Addition of localization analysis by histochemistry using a home-made antibody presented in Fig.1 of the new manuscript

Novel simplified schema is presented as Fig. 9 in the new manuscript.

Overstatements and a speculative schema have been removed from the previous manuscript.

Altogether, I consider that the authors carefully improved and clarified the previous manuscript.

This revised version deserves publication.

Small typos to be corrected.

For instance, in Appendix Table S1. Summary of behavioral analyses of Ldb2 KO mouse

Replace: Barns maze by Barnes maze

]Working deficits by Working deficits

Neurogenesis in the dente gyrus by Neurogenesis in the dentate gyrus

Referee #3 (Remarks for Author):

The manuscript has been vastly improved after the revision. There are some minor errors needing correction, but I recommend publishing this study.

Minor correction

Page 6, "Here, it is of note that most of Neu-positive cells in the layers I and VI were Ldb2-negative." Correct VI to IV.

Page 8, "dopaminergic psychostimulant, than the WT littermates (Fig 3A, upper panel)." "a non-competitive N-methyl-D-aspartate (NMDA) receptor blocker, MK-801 (0.2 mg/kg, s.c.) known as a hallucinogenic drug (Fig 3A, lower panel)". There is no upper and lower panel in Fig 3A. Correct.

Page 8, "It is well known that the amygdala, where significant expression of Ldb2 was seen (Fig 1G)," Fig1G is not correct, should be Fig1E.

Page 11, "Collectively, it is strongly suggested that (1) the lack of Ldb2 protein in LA neurons suppresses Arc-gene expression during fear conditioning," Lack of Ldb2 protein in LA neurons promotes Arc-gene expression. NOT SUPPRESSES.

Page 16, "The ChIP-seq analyses suggested that LDB2/Ldb2 cooperates with EGR transcription factors in the LDB2/Ldb2-driven gene-expression machinery (Fig. 10)." Change Fig.10 to Fig.9.

Our response to the comments raised by the reviewers

Response to Referee #1

Altogether, I consider that the authors carefully improved and clarified the previous manuscript. This revised version deserves publication.

Our response: I would like to deeply thank you for the substantial contribution to improve our manuscript.

Small typos to be corrected.

For instance, in Appendix Table S1. Summary of behavioral analyses of Ldb2 KO mouse

Replace: Barns maze by Barnes maze

]Working deficits by Working deficits

Neurogenesis in the dente gyrus by Neurogenesis in the dentate gyrus

Our response: We have corrected all the typos.

Response to Referee #3

The manuscript has been vastly improved after the revision. There are some minor errors needing correction, but I recommend publishing this study.

Our response: I would like to deeply thank you for the substantial contribution to improve our manuscript.

Minor correction

Page 6, "Here, it is of note that most of Neu-positive cells in the layers I and VI were Ldb2-negative." Correct VI to IV.

Our response: We have corrected it.

Page 8, "dopaminergic psychostimulant, than the WT littermates (Fig 3A, upper panel)." "a non-competitive N-methyl-D-aspartate (NMDA) receptor blocker, MK-801

(0.2 mg/kg, s.c.) known as a hallucinogenic drug (Fig 3A, lower panel)". There is no upper and lower panel in Fig 3A. Correct.

Our response: We have corrected them.

Page 8, "It is well known that the amygdala, where significant expression of Ldb2 was seen (Fig 1G)," Fig1G is not correct, should be Fig1E.

Our response: We have corrected it.

Page 11, "Collectively, it is strongly suggested that (1) the lack of Ldb2 protein in LA neurons suppresses Arc-gene expression during fear conditioning," Lack of Ldb2 protein in LA neurons promotes Arc-gene expression. NOT SUPPRESSES.

Our response: We have corrected the sentence as follows:

Original) a lack of the Ldb2 protein in LA neurons suppresses *Arc*-gene expression during fear conditioning

Revised) the Ldb2 protein in LA neurons suppresses *Arc*-gene expression during fear conditioning

Page 16, "The ChIP-seq analyses suggested that LDB2/Ldb2 cooperates with EGR transcription factors in the LDB2/Ldb2-driven gene-expression machinery (Fig. 10)." Change Fig.10 to Fig.9.

Our response: We have corrected it.

25th Jan 2021

Dear Tetsuo,

We are pleased to inform you that your manuscript is accepted for publication and is now being sent to our publisher to be included in the next available issue of EMBO Molecular Medicine.

Corresponding Author Name: Tetsuo OHNISHI, Takeo YOSHIKAWA, Toshiya Manabe

Manuscript Number: EMM-2020-12574